# Lorentz Local Canonicalization:
# How to Make Any Network Lorentz-Equivariant

**Jonas Spinner**[*1]     **Luigi Favaro**[*2]     **Peter Lippmann**[*3]
**Sebastian Pitz**[1]     **Gerrit Gerhartz**[1]
**Tilman Plehn**[1]     **Fred A. Hamprecht**[3]

[1]ITP, Heidelberg University, Germany,     [2]CP3, UCLouvain, Belgium,

[3]IWR, Heidelberg University, Germany

j.spinner@thphys.uni-heidelberg.de
luigi.favaro@uclouvain.be
peter.lippmann@iwr.uni-heidelberg.de

## Abstract

Lorentz-equivariant neural networks are becoming the leading architectures for high-energy physics. Current implementations rely on specialized layers, limiting architectural choices. We introduce Lorentz Local Canonicalization (LLoCa), a general framework that renders any backbone network exactly Lorentz-equivariant. Using equivariantly predicted local reference frames, we construct LLoCa-transformers and graph networks. We adapt a recent approach for geometric message passing to the non-compact Lorentz group, allowing propagation of space-time tensorial features. Data augmentation emerges from LLoCa as a special choice of reference frame. Our models achieve competitive and state-of-the-art accuracy on relevant particle physics tasks, while being $4\times$ faster and using $10\times$ fewer FLOPs.

## 1 Introduction

Many significant recent discoveries in the natural sciences are enhanced by Machine Learning (ML) [2, 14, 23, 46]. In particular, High Energy Physics (HEP) experiments profit from ML as a vast amount of data is available [12, 22, 25]. For instance, the Large Hadron Collider (LHC) at CERN collects data at rates which are unmatched in the natural sciences, seeking to explain the most fundamental building blocks of nature [13].

The collision between two highly-energetic particles produces a multitude of scattered particles. The physical laws which govern the dynamics of these particles respect Lorentz symmetry, the symmetry group of special relativity. Incorporating the Lorentz symmetry in neural networks, in the form of Lorentz equivariance, has proven to be critical for physics tasks which require data efficiency and high accuracy [7, 21, 39, 42]. However, existing Lorentz-equivariant architectures often rely on task-specific building blocks which impede the general applicability of Lorentz-equivariant networks in the field. Other more versatile equivariant architectures impose prohibitive computational costs, limiting usability at scale, and increasing the energy footprint from training. In this work, we address these limitations by introducing a method that guarantees exact, but also partial and approximate, Lorentz equivariance with minimal additional computational costs. Our approach is agnostic to the architecture of the model, enabling the adaptation of existing graph neural networks (GNNs) and transformers to Lorentz-equivariant neural networks. We make the following contributions:

---

*Equal contribution

39th Conference on Neural Information Processing Systems (NeurIPS 2025).

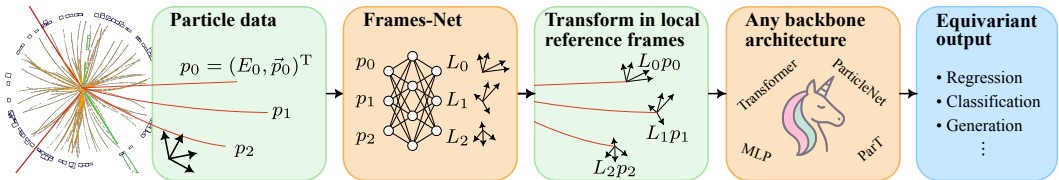

Figure 1: **Lorentz Local Canonicalization (LLoCa) for making any architecture Lorentz-equivariant.** The input consists of a set of particles each associated with an energy and momentum (and possibly other particle features). The particle features are transformed into learned local reference frames, turning them into Lorentz-invariant local features. The local features can then be processed by any backbone architecture to produce exactly Lorentz-equivariant outputs for a variety of possible tasks. Without including additional domain-specific priors, our approach elevates the performance of domain-agnostic models, such as a vanilla transformer, to the SOTA in the field (see e.g. Fig. 3).

- To the best of our knowledge, we introduce the first (local) canonicalization framework for Lorentz-equivariant deep learning, a novel approach that does not rely on specialized layers to achieve internal representations of space-time tensors. We present a novel approach for Lorentz-equivariant prediction of local reference frames and adapt a recently proposed approach [30] for geometric message passing to the non-compact Lorentz group. In particular, we propose a variant of scaled dot-product attention based on the Minkowski product that leverages efficient off-the-shelf attention implementations.

- Our framework is readily applicable to any non-equivariant backbone architectures. It is easy to integrate and can significantly improve the training and inference times over Lorentz-equivariant architectures that use specialized layers.

- In several experiments, we demonstrate the efficacy of exact Lorentz equivariance by achieving state-of-the-art or competitive results using a powerful Lorentz-equivariant transformer and by making several domain-specific networks Lorentz-equivariant.

- Our framework allows for a fair comparison between data augmentation and exact Lorentz equivariance. Our experiments support the superiority of Lorentz-equivariant models when ample training data is available in HEP, while data-augmentation achieves competitive accuracy when training data is scarce.

- Unlike prior equivariant models, our approach incurs only a moderate computational overhead, with a 10–50% increase in FLOPs and a 60–100% increase in training time compared to non-equivariant baselines. Compared to other SOTA Lorentz-equivariant architectures, our models train $4\times$ faster and use $10\times$ fewer FLOPs. Our implementation of LLoCa is publicly available on `https://github.com/heidelberg-hepml/lloca`, and experiments can be reproduced with `https://github.com/heidelberg-hepml/lloca-experiments`.

## 2 Background

**Lorentz group.** The theory of special relativity [19, 35] is built on two postulates: (i) the laws of physics take the same form in every inertial reference frame, and (ii) the speed of light $c$ is identical for all observers. Together they imply that every inertial observer must agree on a single scalar quantity, the spacetime interval $\Delta s^2 = c^2 \Delta t^2 - \Delta \vec{x}^2$. Adopting natural units ($c = 1$), we formalize it with the Minkowski product

$$\langle x, y \rangle = x^0 y^0 - x^1 y^1 - x^2 y^2 - x^3 y^3 . \tag{1}$$

The Minkowski product acts on column four-vectors $x = (x^0, \vec{x})^T \in \mathbb{R}^4$, which can be decomposed into a temporal part $x^0$ and a spatial part $\vec{x} = (x^1, x^2, x^3)^T$. Equation (1) can be compactly written as $x^T g y$ using the Minkowski metric $g = \text{diag}(1, -1, -1, -1)$ [34]. Hence, frame-to-frame transformations can be represented as matrices $\Lambda \in \mathbb{R}^{4 \times 4}$ that preserve the Minkowski product, i.e. that fulfill $\Lambda^T g \Lambda = g$. The inverse transformation is given by $\Lambda^{-1} = g \Lambda^T g$. Lorentz transformations $\Lambda$ act on four-vectors $x$ as $x \to x' = \Lambda x$. The collection of all Lorentz transformations constitutes the Lorentz group $O(1, 3)$. In the following we will focus on the special orthochronous Lorentz

group $\mathrm{SO}^+(1,3)$,[2] which emerges as a subgroup from the constraints that $\det \Lambda = 1$ and $\Lambda_{00} > 0$.

A convenient parametrization of a Lorentz transformation $\Lambda \in \mathrm{SO}^+(1,3)$ is obtained via polar decomposition,[3] which factors the transformation into a purely spatial rotation $R$ and a boost $B$

$$\Lambda = RB = \begin{pmatrix} 1 & \vec{0}^T \\ \vec{0} & \tilde{R} \end{pmatrix} \begin{pmatrix} \gamma & -\gamma\vec{\beta}^T \\ -\gamma\vec{\beta} & I_3 + (\gamma - 1)\frac{\vec{\beta}\vec{\beta}^T}{\vec{\beta}^2} \end{pmatrix}, \qquad \gamma = (1 - \vec{\beta}^2)^{-1/2}. \tag{2}$$

In this decomposition, $R$ leaves the time component of a four-vector untouched while the submatrix $\tilde{R} \in \mathbb{R}^{3\times3}$ rotates its spatial coordinates. Meanwhile, $B$ performs a hyperbolic "rotation" that mixes time with a chosen spatial direction determined by the dimensionless velocity vector $\vec{\beta}$. The Lorentz factor $\gamma = (1 - \vec{\beta}^2)^{-1/2}$ encodes the amount of time-dilation and length-contraction introduced by the boost. Every four-vector $p = (p^0, \vec{p})$, with positive norm $m = \|p\| = (\langle p, p \rangle)^{1/2}$, defines such a boost $B(p)$ with velocity $\vec{\beta} = \vec{p}/p_0$.

**Group representations.** To characterize how transformations of a group $G$ act on elements of a vector space $V$, we use the notion of a group representation. A group representation is a map $\rho$ that assigns an invertible matrix $\rho(g) \in \mathrm{GL}(V)$ to each group element such that $\rho(g_1 g_2) = \rho(g_1)\rho(g_2)$ holds for any pair of group elements $g_1, g_2 \in G$. In this expression, $g_1 g_2$ is the group product and $\rho(g_1)\rho(g_2)$ the matrix product. For instance, four-vectors transform under the 4-dimensional vector representation of the Lorentz group, $x' = \Lambda x$, or in components, $x'^\mu = \sum_\nu \Lambda^\mu{}_\nu x^\nu$. Higher-order tensor representations follow the pattern

$$f'^{\mu_1 \dots \mu_n} = (\rho(\Lambda)f)^{\mu_1 \dots \mu_n} = \sum_{\nu_1 \dots \nu_n} \Lambda^{\mu_1}{}_{\nu_1} \dots \Lambda^{\mu_n}{}_{\nu_n} f^{\nu_1 \dots \nu_n}. \tag{3}$$

A tensor with $n$ indices is said to have order $n$; each index runs over four spacetime directions, so the associated representation acts on a $4^n$-dimensional space.

**Equivariance.** Geometric deep learning takes advantage of the symmetries already present in a problem instead of forcing the model to "rediscover" them from data [11, 15]. An operation $h$ is equivariant to a symmetry group $G$ when the group action commutes with the function, i.e. $h(g \cdot x) = g \cdot h(x)$ for any $g \in G$. Invariance of $h$ emerges as a special case of equivariance if the output is invariant under group actions, $h(g \cdot x) = h(x)$.

**High-energy physics.** In most steps of a HEP analysis pipeline, the data takes the form of a set of *particles*. Each particle is characterized by its discrete particle type as well as its energy $E$ and three-momentum $\vec{p}$ which make up the four-momentum $p = (E, \vec{p})^T \in \mathbb{R}^4$. Four-momenta transform in the vector representation of the Lorentz group. Examples for particle types are fundamental particles like electrons, photons, quarks and gluons as well as reconstructed objects like "jets" [40]. In addition to the particle mass $m = (\langle p, p \rangle)^{1/2}$, particles might be described by extra scalar information such as the electric charge depending on the application. Particle data is typically processed with permutation-equivariant architectures such as graph networks and transformers, with each particle corresponding to one node or token in a fully connected graph.

**Lorentz symmetry breaking.** Although the underlying dynamics of high-energy physics respect the unbroken Lorentz group $\mathrm{SO}^+(1,3)$, the experimental environment in HEP introduces explicit symmetry breaking. The proton beams as well as the detector geometry single out a preferred spatial axis, breaking the symmetry of rotations around axes orthogonal to the beam direction. Further, reconstruction algorithms use the transverse momentum $(p_x^2 + p_y^2)^{1/2}$ to define jets, a quantity which is only invariant under rotations around the beam axis. As a result, many collider observables retain at most a residual $\mathrm{SO}(2)$ symmetry of rotations about the beam axis. In particular, the classification score in Sec. 5.1 is only $\mathrm{SO}(2)$-equivariant, while the regression target in Sec. 5.2 is fully Lorentz-equivariant.

Nevertheless, Lorentz-equivariant networks outperform $\mathrm{SO}(2)$-invariant networks in tasks with

---

[2]"Special" and "orthochronous" means that spatial and temporal reflections are not included in the group.

[3]More generally, a polar decomposition describes the factorization of a square matrix into a hermitian and a unitary part.

partial Lorentz symmetry breaking [7, 21, 39, 42]. This is achieved through a flexible symmetry breaking strategy that preserves Lorentz-equivariant hidden representations. Typically, fixed reference four-vectors such as the global time direction $(1, 0, 0, 0)$ and beam axis $(0, 0, 0, 1)$ are provided as additional inputs, restricting equivariance to transformations that leave these vectors invariant.

## 3 Related work

**Lorentz-equivariant architectures.** Neural networks with exact Lorentz equivariance have emerged as a powerful tool for HEP analyses. A basic approach achieves Lorentz invariance by projecting the particle cloud onto all pairwise Minkowski inner products and processing the results with an MLP [10, 42]. The lack of permutation equivariance in this design is overcome by PELICAN [7], which uses layers that capture all permutation-equivariant mappings on edge features. In order to predict non-scalar quantities such as four-vectors, existing architectures use internal features that transform in Lorentz group representations. LorentzNet [21] uses scalar and vector channels, while CGENN [39] and L-GATr [9, 42] adopt geometric algebra representations, including antisymmetric second-order tensors. CGENN and LorentzNet use message passing, whereas L-GATr relies on self-attention. Unlike these models, LLoCa supports arbitrary representations with fewer architectural constraints and improved efficiency.

**Equivariance by local canonicalization.** An alternative approach to achieve equivariance is to transform the input data into a canonical reference frame. An arbitrary backbone network then acts on the canonicalized input, before a final transformation back to the initial reference frame provides the equivariant output. Such canonicalization can be achieved in two different ways: a) via global canonicalization [24], which uses a global reference frame for the whole point cloud, which is not used in our method, or b) via local canonicalization, where each point in the point cloud is equipped with its own reference frame. In contrast to global canonicalization, local canonicalization facilitates that similar local substructures will yield similar local features. Several methods have been proposed to find a global canonical orientation for 3D point cloud data [6, 28, 49, 50], or a local canonicalization for the compact group of 3D rotations and reflections [17, 31, 47]. However, to the best of our knowledge, there exist no equivalents for the non-compact Lorentz group. As shown in [30], properly transforming tensorial objects between different local reference frames during message passing substantially increases the expressivity of the architecture. We extend their solution to the Lorentz group. While the mentioned approaches use one global reference frame or one local reference frame per node, it is possible and in some cases advantageous to extend the framework of canonicalization to multiple different reference frames (per entity), as presented in the form of frame averaging in [18, 29, 36]. Extending these methods to our framework could be a promising direction for future research.

**Scaling of exact equivariance vs. data augmentation.** Recent work has begun to probe approximate symmetries in neural networks [27] and to question whether exact equivariance improves data scaling laws of neural networks [8, 30]. Building on local canonicalization, our framework enables a controlled comparison: it allows us to evaluate an exactly Lorentz-equivariant model and an equally engineered, non-equivariant counterpart trained with data augmentation on equal footing. We conduct several experiments to investigate how both types of models scale with the amount of training data.

## 4 Methods

The general idea of our framework is the following (cf. Fig. 1): for every particle, or node, we predict a local reference frame which transforms equivariantly under general Lorentz transformations. Then, we express the particle features in the predicted local reference frames, therefore transforming them into Lorentz-invariant features which can be processed by any backbone network. In the end, we obtain a Lorentz-equivariant prediction by transforming the space-time objects back to the global reference frame. In its minimal version, only local scalar information is exchanged between nodes, a constraint that significantly limits the expressivity of the architecture. We instead propose to group particle features into tensor representations to enable the exchange of tensorial space-time messages between different reference frames, extending the method proposed by Lippmann, Gerhartz et al. [20, 30].

### 4.1 Local reference frames

A local reference frame $L$ is constructed from a set of four four-vectors $u_0, u_1, u_2, u_3$ ("vierbein") that form an orthonormal basis in Minkowski space, i.e. they satisfy the condition

$$\sum_{\mu,\nu} u_a{}^\mu u_b{}^\nu g_{\mu\nu} = g_{ab} . \tag{4}$$

The transformation behavior of four-vectors $u_i \to \Lambda u_i$ implies that the local reference frames $L$ transform as $L \to L\Lambda^{-1}$ if constructed from row vectors in the following way[4]

$$L = \begin{pmatrix} \text{---}\ u_0^T g\ \text{---} \\ \text{---}\ u_1^T g\ \text{---} \\ \text{---}\ u_2^T g\ \text{---} \\ \text{---}\ u_3^T g\ \text{---} \end{pmatrix} \quad \overset{\Lambda}{\to} \quad L' = \begin{pmatrix} \text{---}\ u_0^T \Lambda^T g\ \text{---} \\ \text{---}\ u_1^T \Lambda^T g\ \text{---} \\ \text{---}\ u_2^T \Lambda^T g\ \text{---} \\ \text{---}\ u_3^T \Lambda^T g\ \text{---} \end{pmatrix} = L\Lambda^{-1}. \tag{5}$$

In the final step, we insert two Minkowski metrics as $I_4 = gg$, allowing us to identify the inverse transformation $\Lambda^{-1} = g\Lambda^T g$. With this relation, the orthonormality condition (4) becomes $LgL^T = g$, the defining property of Lorentz transformations. In Sec. 4.5, we explicitly construct local reference frames that satisfy both conditions introduced above. When working with sets of particles, a separate local reference frame can be assigned to each particle, see Fig. 1.

### 4.2 Local canonicalization

Local four-vectors $x_L = Lx$ are defined by transforming global four-vectors $x$ into the local reference frame $L$. Thanks to the transformation rule $L \to L\Lambda^{-1}$, these local vectors remain invariant under global Lorentz transformations $\Lambda$, i.e. $x_L = Lx \to x'_L = L\Lambda^{-1}\Lambda x = Lx = x_L$. This property readily generalizes to general Lorentz tensors $f$ (cf. Eq. (3)), which transform as $f \to f' = \rho(\Lambda)f$. For features in the local reference frame $f_L = \rho(L)f$ we find

$$f_L \overset{\Lambda}{\to} f'_L = \rho(L')f' = \rho(L\Lambda^{-1})\rho(\Lambda)f = \rho(L\Lambda^{-1}\Lambda)f = \rho(L)f = f_L. \tag{6}$$

The local particle features $f_L$ can be processed with any backbone architecture without violating their Lorentz invariance, cf. Fig. 1. Finally, output features $y$ in the global reference frame are extracted using the inverse transformation $y = \rho(L^{-1})f_L$. These output features $y$ are equivariant under Lorentz transformations

$$y \overset{\Lambda}{\to} y' = \rho(L'^{-1})f'_L = \rho(\Lambda L^{-1})f_L = \rho(\Lambda)\rho(L^{-1})f_L = \rho(\Lambda)y. \tag{7}$$

### 4.3 Tensorial messages between local reference frames

Consider a system of $N$ particles, where a message is sent from particle $j$ to particle $i$. The message $m_{j,L_j}$ in the local reference frame $L_j$ is Lorentz-invariant, as it is constructed from the invariant particle features $f_{L_j}$. To communicate this message to particle $i$ in reference frame $L_i$, we apply the reference frame transformation matrix $L_i L_j^{-1}$, which gives the transformed message:

$$m_{j,L_i} = \rho(L_i L_j^{-1})m_{j,L_j}. \tag{8}$$

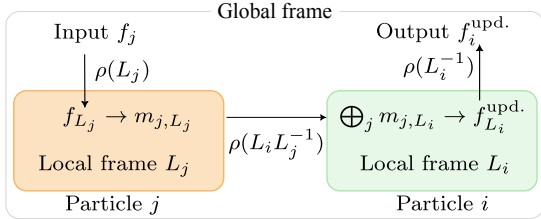

Figure 2: **Tensorial message passing in LLoCa.** Lorentz-invariant local messages are transformed between local reference frames.

The message representation $\rho$ is a hyperparameter that can be chosen according to the problem. Our implementation supports general tensor representations following Eq. (3), though in practice, we find an equal mix of scalar and vector representations sufficient for our applications. Regardless of the chosen representation, the received message $m_{j,L_i}$ remains invariant since the transformation matrix $L_i L_j^{-1}$ is invariant:

$$L_i L_j^{-1} \overset{\Lambda}{\to} L'_i L_j'^{-1} = L_i \Lambda^{-1}(L_j \Lambda^{-1})^{-1} = L_i \Lambda^{-1}\Lambda L_j^{-1} = L_i L_j^{-1}. \tag{9}$$

---

[4]The object $u^\flat = u^T g$ is called the covector of the vector $u$, with the transformation behavior $u'^\flat = u^\flat \Lambda^{-1}$.

This formalism can be easily integrated into any message passing paradigm. Writing $\phi$ and $\psi$ for unconstrained neural networks and $\bigoplus$ for a permutation-invariant aggregation operation, the updated features for particle $i$ can be written as:

$$f_{L_i}^{\text{updated}} = \psi\left( f_{L_i}, \bigoplus_{j=1}^N \phi\big(\rho(L_i\, L_j^{-1})\, m_{j,L_j}\big) \right). \tag{10}$$

We extend this framework proposed by Lippmann, Gerhartz et al. [30] to the Lorentz group, see Fig. 2.

**Tensorial scaled dot-product attention.** A particularly prominent example of message passing is scaled dot-product attention [44]. We obtain scaled dot-product attention with tensorial message passing as a special case of Eq. (10)

$$f_{L_i}^{\text{updated}} = \sum_{j=1}^N \text{softmax}\left( \frac{1}{\sqrt{d}} \Big\langle q_{L_i}, \rho(L_iL_j^{-1})k_{L_j} \Big\rangle \right)\, \rho(L_iL_j^{-1})\, v_{L_j}. \tag{11}$$

The objects $q_{L_i}, k_{L_j}, v_{L_j}$ are the $d$-dimensional queries, keys and values, each computed in the respective local reference frame from the respective local node features $f_{L_i}, f_{L_j}$. The softmax normalizes the attention weights over all sending nodes. The Minkowski product $\langle \cdot, \cdot \rangle$ reduces to the Euclidean product if the queries, keys and values are assigned to scalar representations $\rho$. For higher-order representations, we find that Minkowski attention outperforms Euclidean attention, see App. E. Since the Minkowski product is Lorentz-invariant,

$$\langle q_{L_i}, \rho(L_iL_j^{-1})k_{L_j}\rangle = \langle \rho(L_i^{-1})q_{L_i}, \rho(L_i^{-1})\rho(L_iL_j^{-1})k_{L_j}\rangle = \langle \rho(L_i^{-1})q_{L_i}, \rho(L_j^{-1})k_{L_j}\rangle. \tag{12}$$

That is, keys and queries can simply be transformed into the global frame of reference, in which all Minkowski products can be efficiently evaluated. This facilitates the use of highly optimized scaled dot-product attention implementations [16].

### 4.4 Relation to data augmentation

Architectures with local canonicalization achieve Lorentz equivariance if the local reference frames satisfy Eq. (5), imposing strong constraints on their construction. Non-equivariant networks emerge as a special case where the local reference frames are the identity, $L = 1$. Data augmentation also emerges as a special case, where one global reference frame is drawn from a given distribution of Lorentz transformations. Implementing non-equivariant networks and data augmentation as special cases of equivariant networks enables a fair comparison between these three choices.

### 4.5 Constructing local reference frames

We now describe how to construct local reference frames for a set of particles, the typical data type in high-energy physics. Each local reference frame $L$ has to satisfy the Lorentz group condition $L^T g L = g$ and transform as $L \to L' = L\Lambda^{-1}$ under Lorentz transformations $\Lambda$. The local reference frames are constructed in two steps: first we use a simple Lorentz-equivariant architecture to predict three four-vectors for each particle. The four-vectors are then used to construct the local reference frames $L$ following a deterministic algorithm.

In a system of $N$ particles, each particle is described by its four-momentum $p_i$ and additional scalar attributes $s_i$, such as the particle type. For each particle $i$, the three four-vectors are predicted as

$$v_{i,k} = \sum_{j=1}^N \text{softmax}\Big( \varphi_k(s_i, s_j, \langle p_i, p_j\rangle) \Big) \frac{p_i + p_j}{\|p_i + p_j\| + \epsilon} \quad \text{for } k = 0, 1, 2. \tag{13}$$

This operation is Lorentz-equivariant because it constructs four-vectors as a linear combination of four-vectors with scalar coefficients [45]. The function $\varphi$ is an MLP with three output channels that operates on Lorentz scalars. Empirically, we find that a network $\varphi$ that is significantly smaller than the backbone network is already sufficient, see App. D. To ensure positive and normalized weights, a softmax is applied across the weights of all sending nodes and the four-momenta are rescaled by their norm. This constrains the resulting vectors $v_{i,k}$ to have positive inner products $\langle v_{i,k}, v_{i,k}\rangle > 0$, given that the input particles have positive inner products and positive energy $p_i^0 > 0$. We observe that the softmax operation significantly improves numerical stability during the subsequent orthonormalization procedure.

Indeed, large boosts $B$ may lead to numerical instabilities. However, although the particles $p_i$ are highly boosted, we find that the predicted vectors $v_{i,k}$ typically are not, resulting in stable training. Appendix D.1 holds a detailed discussion on the numerical stability of the local frame prediction. In tasks where symmetry breaking effects reduce full Lorentz symmetry to a subgroup, we incorporate reference particles (cf. Sec. 2) which are needed only for the local frame prediction.

For each particle $i$, the local reference frame $L_i$ is constructed from the three four-vectors $v_{i,k}, k = 0, 1, 2$ using polar decomposition, cf. Eq. (2). For readability, we omit the particle index $i$ from here on. As outlined in Alg. 1, the boost $B(v_0)$ is built from $v_0$, while the rotation $R$ is derived from $v_1$ and $v_2$. These two vectors

---

**Algorithm 1** Local reference frames via polar decomposition

---

**Require:** $v_0, v_1, v_2 \in \mathbb{R}^4$ with $v'_i = \Lambda v_i$, $\langle v_0, v_0 \rangle > 0$
**Ensure:** $L^T g L = g$ and $L \to L' = L\Lambda^{-1}$
1: $B \leftarrow B(v_0)$ using Eq. (2)
2: $w_k \leftarrow B v_k$ for $k = 1, 2$
3: $\vec{u}_1, \vec{u}_2, \vec{u}_3 \leftarrow \text{GramSchmidt}(\vec{w}_1, \vec{w}_2)$
4: $\tilde{R} \leftarrow (\vec{u}_1, \vec{u}_2, \vec{u}_3)^T$
5: $R \leftarrow \begin{pmatrix} 1 & \vec{0}^T \\ \vec{0} & \tilde{R} \end{pmatrix}$
6: $L \leftarrow RB$

---

are first transformed with the same boost and then orthonormalized via the Gram-Schmidt algorithm. Alg. 1 is equivalent to a Gram-Schmidt algorithm in Minkowski space; see App. C for details and a proof that $L$ satisfies the transformation rule $L \to L\Lambda^{-1}$. The above construction contains equivariantly predicted rotations for SO(3)-equivariant architectures as a special case with $v_0 = (1, 0, 0, 0)$, though this restricted variant yields inferior performance, see App. E. In App. B we describe how the LLoCa framework can be extended to the Poincaré group and $O(1, d)$.

**Limitations.** The Gram-Schmidt orthogonalization requires linear independence of $v_1$ and $v_2$. In practice, we find this holds reliably for systems with $N \geq 3$ particles as we carefully address the numerical stability of the local frames prediction (see App. D.1). For systems with $N < 3$ particles, it is not possible to construct equivariant local frames from the information contained in the input set of particles. We handle this very special case by sampling the missing spatial directions $\vec{w}_1$ and $\vec{w}_2$ (cf. Alg. 1), randomly from an SO(3)-invariant distribution, at the cost of formally breaking Lorentz-equivariance to SO(3) or SO(2)-equivariance. In situations with partial Lorentz symmetry breaking, see Sec. 2, the reference particles encoding the time and beam direction can lift the number of input vectors $p_i$ to 3 or higher.

## 5 Experiments

We now demonstrate the effectiveness of Lorentz Local Canonicalization (LLoCa) for a range of different architectures on two relevant tasks in HEP. We start from the classification of jets, or "jet tagging". Then, we present extensive studies on QFT amplitude regression.

### 5.1 Jet tagging

Jet tagging is the identification of the mother particle which initiated the spray of hadrons, a "jet", from a set of reconstructed particles. This task plays a key role in HEP workflows, where even marginal gains in classification performance lead to purer datasets – ultimately saving experimental resources. For this application the big-data regime is most interesting, because simulating training data is cheap. Therefore, we use the JetClass dataset [38], a modern benchmark dataset that contains 125M jets divided into 10 classes.

The ParticleNet [37] and ParT [38] networks are established jet tagging architectures in the high-energy physics community. ParticleNet uses dynamic graph convolutions, and ParT is a transformer with class attention [43] and a learnable attention bias based on Lorentz-invariant edge attributes. We use the LLoCa framework to make both architectures Lorentz-equivariant with minimal adaptations to the official implementation, see Tab. 1. Further, we compare against a vanilla transformer and its LLoCa adaptation (cf. Sec. 4.3), see App. D for details.

Without any hyperparameter tuning, the Lorentz-equivariant models based on LLoCa consistently outperform their non-equivariant counterparts, at the cost of some extra training time and FLOPs (see Tab. 1). Interestingly, LLoCA can elevate the performance of the domain-agnostic transformer to the performance of the domain-specific ParT architecture. This indicates that LLoCa provides an effective inductive bias, without the need for specialized architectural designs.

Table 1: **LLoCa consistently improves the performance of non-equivariant architectures.** For classification on the JetClass dataset we compare accuracy, area under the ROC curve (AUC) as well as training time on a H100 GPU and FLOPs for a forward pass with batch size 1 for the JetClass dataset from [38]. For our models, accuracy and AUC metrics are significant up to the last digit. LLoCa improves domain-specific architectures and elevates a vanilla transformer to competitive accuracy (* indicates Lorentz equivariance). For more metrics, see Tab. 5.

| Model | Accuracy (↑) | AUC (↑) | Time | FLOPs |
|---|---|---|---|---|
| PFN [26] | 0.772 | 0.9714 | 3h | 3M |
| P-CNN [41] | 0.809 | 0.9789 | 3h | 12M |
| LorentzNet [21] | 0.847 | 0.9856 | 64h | 676M |
| MIParT-L [48] | 0.861 | 0.9878 | 43h | 225M |
| L-GATr* [9] | **0.866** | **0.9885** | 166h | 2060M |
| ParticleNet [37] | 0.844 | 0.9849 | 25h | 413M |
| LLoCa-ParticleNet* | 0.845 | 0.9852 | 43h | 517M |
| ParT [38] | 0.861 | 0.9877 | 33h | 211M |
| LLoCa-ParT* | 0.864 | 0.9882 | 66h | 315M |
| Transformer | 0.855 | 0.9867 | 15h | 210M |
| LLoCa-Transformer* | 0.864 | 0.9882 | 31h | 301M |

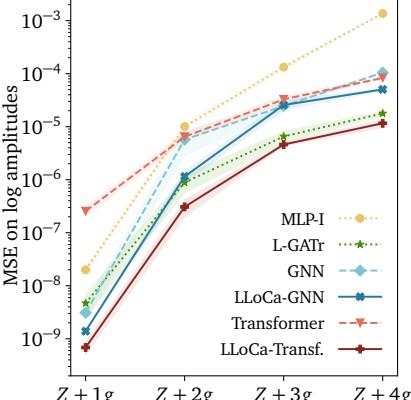

| Model | MSE$\times 10^{-5}$ (↓) | FLOPs | Time |
|---|---|---|---|
| MLP-I [10, 42] | 137.0 ± 2 | 0.1M | 0.4h |
| L-GATr [42] | 1.8 ± 0.2 | 528.0M | 8.3h |
| GNN | 10.5 ± 0.2 | 20.7M | 0.9h |
| LLoCa-GNN | 5.0 ± 0.2 | 22.3M | 1.5h |
| Transformer | 8.3 ± 0.3 | 14.9M | 1.3h |
| LLoCa-Transformer | **1.2** ± 0.2 | 16.3M | 2.1h |

Figure 3: **LLoCa surpasses SOTA performance while being $4\times$ faster.** The collisions of two particles produces a single $Z$ boson and $n = 1, 2, 3, 4$ gluons $g$ (x-ticks $Z + ng$). LLoCa significantly improves the prediction of interaction amplitudes of the non-equivariant GNN and transformer. Left: Our LLoCa-Transformer achieves state-of-the-art results over all four multiplicities. Right: Accuracy and compute for $Z + 4g$. The LLoCa-Transformer uses a tenth of the FLOPs and a fourth of the training time relative to the second most accurate model. Uncertainties are standard deviations over three runs. See App. D for more details.

## 5.2 QFT amplitude regression

In collisions of fundamental particles, the occurrence of a specific final state is governed by a probability. This probability is a function of the four-momenta of the incoming and outgoing particles, as well as their particle types. Using the framework of Quantum Field Theory (QFT), we can calculate this probability or "amplitude" in a perturbative expansion. Neural surrogates [3–5, 10, 32, 33, 42] can greatly speed up this process, since the factorial growth of possible interactions with increasing particle number often makes exact evaluation unfeasible. The exact Lorentz invariance of amplitudes makes them a prime candidate for a LLoCa surrogate model.

We follow [42] and benchmark our Lorentz Local Canonicalization (LLoCa) approach on the process $q\bar{q} \rightarrow Z + ng$, the production of a $Z$ boson with $n = 1, \ldots, 4$ additional gluons from a quark-antiquark pair. The surrogates are trained for each value of $n$ separately. We evaluate the neural surrogates in terms of the Mean Squared Error (MSE) between the predictions and the ground truth of the logarithmic amplitudes. Architectures and training details are described in App. D.

LLoCa allows us to upgrade any non-equivariant baseline and directly study the benefits of built-in

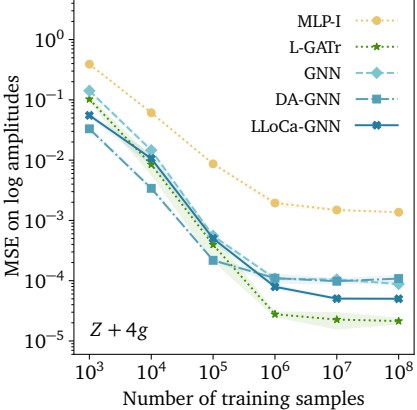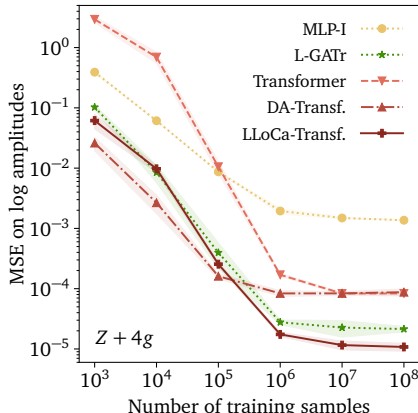

Figure 4: **Data efficiency of LLoCa networks.** Lorentz-equivariant amplitude regression for $Z + 4g$ production is more data efficient in the large data regime for both graph networks (left) and transformers (right). Perhaps surprisingly, data augmentation (DA) outperforms equivariance for small datasets.

Lorentz equivariance. To that end, we construct LLoCa adaptations of a baseline graph neural network (GNN) and a vanilla transformer as described in Sec. 4.3. For comparison, we consider a simple MLP acting on Lorentz-invariants (MLP-I) [10] and the performant L-GATr [42]. LLoCa consistently improves the accuracy of non-equivariant architectures, see Fig. 3. Crucially, without including additional domain-specific priors, our LLoCa-Transformer achieves state-of-the-art-performance, outperforming the significantly more expensive L-GATr architecture over the entire range of multiplicities.

**Message representations.** Beyond the improvements obtained through exact Lorentz equivariance, we have identified the tensorial message passing in LLoCa (cf. Sec. 4.3) as a crucial factor for expressive message passing based on local canonicalization. We compare the performance of the LLoCa-transformer with different message representations in the 16-dimensional attention head, see Tab. 2. We compare 16 scalars against four 4-dimensional vectors, a single 16-dimensional second-order tensor representations as well as our default of eight scalars combined with two four-vectors. All tensorial representations significantly out-

Table 2: **Effectiveness of tensorial messages.** LLoCa-Transformer with different hidden tensor representations on $Z + 4g$ QFT amplitude regression.

| Method | MSE ($\times 10^{-5}$) |
|---|---|
| Non-equivariant | $8.3 \pm 0.5$ |
| Global canonical. | $4.4 \pm 1.0$ |
| LLoCa (16 scalars) | $40 \quad \pm 4$ |
| LLoCa (single 2-tensor) | $2.0 \pm 0.4$ |
| LLoCa (4 vectors) | $1.4 \pm 0.2$ |
| LLoCa (8 scalars, 2 vectors) | $\mathbf{1.0} \pm 0.1$ |

perform the scalar message passing. For reference, we include the non-equivariant transformer and the same model with global canonicalization (one shared learned reference frame to every particle).

**Lorentz equivariance at scale.** Lorentz equivariance introduces a strong inductive bias in neural networks, typically at the cost of extra compute. Data augmentation provides a cheap alternative which aims at learning approximate Lorentz equivariance directly from data. LLoCa includes data augmentation as a special case, allowing us to directly compare exact Lorentz equivariance (LLoCa) with data augmentation (DA) using the same backbone architecture and training parameters. For this purpose, we focus on the most complicated process $Z + 4g$ and train surrogate models with different fractions of the training dataset, see Fig. 4. Augmented trainings outperform the non-equivariant baseline for small training datasets due to the symmetry group information encoded in data augmentations, but the two approaches agree for large dataset sizes where model expressivity becomes the limiting factor. The Lorentz-equivariant models surpass the competition in this big-data regime due to their increased expressivity at fixed parameter count. Interestingly, we observe that trainings with data augmentation outperform even the Lorentz-equivariant models for small training dataset sizes. We include extensive ablation studies on the method used for local canonicalization, tensorial message representations, and data augmentation in App. E.

## 5.3 Computational efficiency

While Lorentz-equivariant architectures offer valuable inductive biases, they often come with significant computational overhead compared to the corresponding non-equivariant architectures. To evaluate this trade-off, we report both nominal FLOPs – which capture architectural complexity – and empirical training time, which also reflects implementation efficiency, in Tab. 1 and Fig. 3. Our LLoCa networks introduce a modest increase in computational cost, with FLOPs rising by 10–50% and training time by 60–100%, depending on the task and architecture. The computational overhead of LLoCa networks is split about evenly between (i) the initial prediction of local frames and (ii) the frame-to-frame transformations in tensorial message passing. A more efficient implementation is likely to further reduce both sources of training-time overhead within LLoCa networks. In contrast, the state-of-the-art L-GATr architecture [42] incurs significantly higher costs, with a $10\times$ increase in FLOPs and a $4\times$ increase in training time relative to our LLoCa-Transformer, see App. E.2.

## 6 Conclusion

Particle physics provides ample data and studies systems that exhibit Lorentz symmetry. Neural networks that respect Lorentz symmetry have emerged as essential tools in accurately analyzing the data and modeling the underlying physics. However, existing Lorentz-equivariant architectures introduce large computational costs and rely on specialized building blocks, which limits the architectural design space and hinders the transfer of deep learning progress from other modalities. We address these issues by introducing Lorentz Local Canonicalization (LLoCa), a novel framework that can make any off-the-shelf architecture Lorentz-equivariant. In several experiments, we have demonstrated the effectiveness of LLoCa by achieving state-of-the-art results even with domain-agnostic backbone architectures. Furthermore, our approach significantly improves established domain-specific but non-equivariant architectures, without any hyperparameter tuning. While the LLoCa framework introduces a computational overhead, our LLoCa models are still significantly faster at training and inference than other SOTA Lorentz-equivariant networks. We hope that LLoCa facilitates the design of novel Lorentz-equivariant architectures and helps to bridge innovations between machine learning in particle physics and other domains.

**Acknowledgements**

The authors thank Víctor Bresó for generating the amplitude regression datasets. They are also grateful to Huilin Qu for assistance with the JetClass dataset and for support with the ParticleNet and ParT implementations. Further thanks go to Johann Brehmer, Víctor Bresó, Jesse Thaler, and Henning Bahl for valuable discussions and insightful feedback on the manuscript.

The authors acknowledge support by the state of Baden-Württemberg through bwHPC and the German Research Foundation (DFG) through the grants INST 35/1597-1 FUGG and INST 39/1232-1 FUGG. Computational resources have been provided by the supercomputing facilities of the Université catholique de Louvain (CISM/UCL) and the Consortium des Équipements de Calcul Intensif en Fédération Wallonie Bruxelles (CÉCI) funded by the Fond de la Recherche Scientifique de Belgique (F.R.S.-FNRS) under convention 2.5020.11 and by the Walloon Region. The present research benefited from computational resources made available on Lucia, the Tier-1 supercomputer of the Walloon Region, infrastructure funded by the Walloon Region under the grant agreement n°1910247.

J.S. and T.P. are supported by the Baden-Württemberg-Stiftung through the program *Internationale Spitzenforschung*, project *Uncertainties — Teaching AI its Limits* (BWST_IF2020-010), the Deutsche Forschungsgemeinschaft (DFG, German Research Foundation) under grant 396021762 – TRR 257 *Particle Physics Phenomenology after the Higgs Discovery*. J.S. is funded by the Carl-Zeiss-Stiftung through the project *Model-Based AI: Physical Models and Deep Learning for Imaging and Cancer Treatment*. L.F. is supported by the Fonds de la Recherche Scientifique - FNRS under Grant No. 4.4503.16. P.L. and F.A.H. acknowledge funding by Deutsche Forschungsgemeinschft (DFG, German Research Foundation) – Projektnummer 240245660 - SFB 1129.

This work is supported by Deutsche Forschungsgemeinschaft (DFG) under Germany's Excellence Strategy EXC-2181/1 - 390900948 (the Heidelberg STRUCTURES Excellence Cluster).

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

## A  Tensor representation

The tensor representation is introduced in Eq. (3). The tensor representation is a group representation of the Lorentz group $O(1, 3)$ and is used in this paper to define the transformation behavior of tensorial space-time features when transformed in or out of local reference frames.

**Proof representation property.**  The representation property follows from the fact that the $4 \times 4$ Lorentz transformation matrices itself form a representation. For any $\Lambda_1, \Lambda_2 \in SO^+(1, 3)$ we have

$$
\begin{aligned}
[\rho(\Lambda_1)\rho(\Lambda_2)T]^{\mu_1 \cdots \mu_n} &= \Lambda_1^{\mu_1}{}_{\rho_1} \cdots \Lambda_1^{\mu_n}{}_{\rho_n} \Lambda_2^{\rho_1}{}_{\nu_1} \cdots \Lambda_2^{\rho_n}{}_{\nu_n} T^{\nu_1 \cdots \nu_n} \\
&= (\Lambda_1\Lambda_2)^{\mu_1}{}_{\nu_1} \cdots (\Lambda_1\Lambda_2)^{\mu_n}{}_{\nu_n} = [\rho(\Lambda_1\Lambda_2)T]^{\mu_1 \cdots \mu_n}
\end{aligned}
\tag{14}
$$

**Representation hidden features.**  The input and output representations are determined by the input data and the prediction task. Contrarily, the internal representations are chosen as hyperparameters. In our framework, we choose a direct sum of tensor representations, which again forms a representation. A feature $f$ will transform under $\rho_f = \rho_1 \oplus \ldots \oplus \rho_k$ with a block diagonal matrix:

$$
\rho_f(\Lambda)f = \begin{pmatrix} \rho_1(\Lambda) & & \\ & \ddots & \\ & & \rho_k(\Lambda) \end{pmatrix} f
\tag{15}
$$

So that the composition of scalar, vectorial and tensorial features can be chosen freely. The feature dimension of $f$ will be given by $\dim(f) = \dim(\rho_1) + \ldots + \dim(\rho_k)$.

## B  Extension of LLoCa to $O(1, d)$ and the Poincaré group

First, note that LLoCa does not cover the full Lorentz group including parity and time reversal, but only the fully-connected subgroup denoted by $SO^+(1, 3)$. An extension to $O(1, 3)$ requires to also predict extra parity and time reversal transformations, similar to what has been done in [30] for extending local canonicalization from $SO(3)$ to $O(3)$. Afterwards, extending LLoCa to $O(1, d)$ is straightforward. Using Eq. (13), one may predict $d$ many $d + 1$-dimensional vectors. Then, a trivial extension of Alg. 1 with $d$-dimensional Gram-Schmidt algorithm for the spatial rotation can be used to predict orthonormal local frames for canonicalization w.r.t. $O(1, d)$. Furthermore, the tensorial message passing based on the tensor representations defined by Eq. (3) can be extended directly to the case of $O(1, d)$.

In addition, invariance w.r.t. translations could be obtained in the LLoCa framework simply by operating only on differences of four-vectors. In all our experiments, the input is provided in momentum space where translation equivariance/invariance is not preferable. However, an extension to the Poincaré group is possible by modifying Eq. (13) to

$$
v_{i,k} = \sum_{j=1}^{N} \text{softmax}(\varphi_k(s_i, s_j, \|x_i - x_j\|))(x_i - x_j),
\tag{16}
$$

where we assume to work in position space with four-vectors $x_i$. One possible challenge may be to ensure that at least one of the predicted $v_{i,k}$ has a positive norm from which one can construct the boost $B$, cf. Alg. 1.

## C  Background information on the construction of local reference frames

### C.1  Gram-Schmidt algorithm in Minkowski space

In this section we discuss how to construct reference frames $L$ that satisfy the transformation behaviour $L \to L\Lambda^{-1}$ of Eq. (5). This can be achieved via Gram-Schmidt orthonormalization algorithm adapted for Minkowski vectors. Similar to Alg. 1, this algorithm starts from three Lorentz vectors $v_0, v_1, v_2$ and returns a local reference frame $L$. While, in Alg. 1, $L$ is decomposed into a rotation and a boost, here we construct $L$ directly from a set of four orthonormal vectors, as illustrated in Eq. (5).

Starting from three Lorentz vectors $v_0, v_1, v_2$, we apply the following algorithm to obtain four four-

vectors $u_0, u_1, u_2, u_3$ that satisfy Eq. (4)

$$
\begin{aligned}
\text{norm}_4(v) &:= \frac{v}{\|v\| + \epsilon}, \\
u_0 &= \text{norm}_4(v_0), \\
u_1 &= \text{norm}_4\left(v_1 - u_0 \frac{\langle v_1, u_0 \rangle}{\langle u_0, u_0 \rangle}\right), \\
u_2 &= \text{norm}_4\left(v_2 - u_0 \frac{\langle v_2, u_0 \rangle}{\langle u_0, u_0 \rangle} - u_1 \frac{\langle v_2, u_1 \rangle}{\langle u_1, u_1 \rangle}\right), \\
u_3^\mu &= \sum_{\nu, \rho, \sigma, \kappa} g^{\mu\nu} \epsilon_{\nu\rho\sigma\kappa} u_0^\rho u_1^\sigma u_2^\kappa.
\end{aligned}
\tag{17}
$$

We use $\epsilon = 10^{-15}$ and the absolute value of the Lorentz inner product $\|x\| = \sqrt{|\langle x, x \rangle|}$. The last line uses the Minkowski index notation to construct $u_3$ using the totally antisymmetric tensor $\epsilon_{\mu\nu\rho\sigma}$, the extension of the cross product to Minkowski space. This tensor is defined as $\epsilon_{0123} = 1$, and it flips sign under permutation of any pair of indices and is zero otherwise.

We identified two advantages to the polar decomposition approach (PD) described in Alg. 1 compared to the Gram-Schmidt algorithm in Minkowski space ($\text{GS}^4$) described in Eq. 17 above. First, the PD has the conceptional advantage of explicitly separating the boost and rotation. This allows to easily study properties of the boost and rotation parts independently, to identify the special cases of a pure boost and a pure rotation, and, if necessary, to constrain the magnitude of the boost vector $v_0$. Second, we find that the PD approach of boosting into a reference frame typical for the particle set and then orthonormalising two vectors in that frame is numerically more stable than the $\text{GS}^4$ approach of directly orthonormalising a set of three vectors, although the approaches are formally equivalent, see App. C.2.

When constructing the local reference frame $L$ from this orthonormal set of vectors $u_0, u_1, u_2, u_3$, it is important to additionally transform them into covectors $u_k^T g$, where $u_k$ denote the vectors constructed in the algorithm above.

## C.2 Proof that polar decomposition via Alg. 1 is equivalent to a Gram-Schmidt algorithm in Minkowski space

Let us first consider the Gram-Schmidt algorithm in Minkowski space (denoted by $\text{GS}^4$). As for Alg. 1 we start from the set of four-vectors $v_k, k = 0, 1, 2$ as predicted by Eq. (13), assuming $\langle v_0, v_0 \rangle > 0$. Now, we use the Gram-Schmidt algorithm in Minkowski space described in Eq. (17), to orthonormalize the three vectors: $\{u_k\}_{k=0,1,2,3} = \text{GS}^4(v_0, v_1, v_2)$. Then, the transformation matrix constructed from these orthonormal vectors is

$$
L^{\text{GS}} = \begin{pmatrix} \text{---} \, u_0^T g \, \text{---} \\ \text{---} \, u_1^T g \, \text{---} \\ \text{---} \, u_2^T g \, \text{---} \\ \text{---} \, u_3^T g \, \text{---} \end{pmatrix}
\tag{18}
$$

Equation (18) has the correct transformation behavior $L \to L\Lambda^{-1}$ as shown in Eq. (5) since $\{u_k\}_{k=0,1,2,3}$ are also four-vectors due to the Lorentz-equivariance of $\text{GS}^4$. According to the polar decomposition, we can decompose a proper Lorentz transformation into a pure rotation $R$ and a general boost $B$. By realizing that

$$
L = RB = \begin{pmatrix} 1 & 0 \\ 0 & \tilde{R} \end{pmatrix} \begin{pmatrix} \gamma & -\gamma\vec{\beta}^{\text{T}} \\ -\gamma\vec{\beta} & \dots \end{pmatrix} = \begin{pmatrix} \gamma & -\gamma\vec{\beta}^{\text{T}} \\ \dots & \dots \end{pmatrix},
\tag{19}
$$

we can identify the boost part of a Lorentz transformation from the first row, i.e. if we consider $L^{\text{GS}} = R^{\text{GS}} B^{\text{GS}}$

$$
B^{\text{GS}} = B(\vec{\beta} = -\vec{u}_0/u_0^0) =: B(u_0).
\tag{20}
$$

Also note that $u_0 = v_0/\langle v_0, v_0 \rangle$ due to the design of $\mathrm{GS}^4$, therefore $B(u_0) = B(v_0)$. Consequently, the pure rotation $R^{\mathrm{GS}}$ is given by

$$R^{\mathrm{GS}} = L^{\mathrm{GS}}(B^{\mathrm{GS}})^{-1} = \begin{pmatrix} \text{---} \ u_0^T g B(v_0)^{-1} \ \text{---} \\ \text{---} \ u_1^T g B(v_0)^{-1} \ \text{---} \\ \text{---} \ u_2^T g B(v_0)^{-1} \ \text{---} \\ \text{---} \ u_3^T g B(v_0)^{-1} \ \text{---} \end{pmatrix}. \tag{21}$$

We now show that the local frame $L^{\mathrm{PD}}$ obtained from polar decomposition (PD) via Alg. 1 is equal to $L^{\mathrm{GS}}$. Starting from the same set of four-vectors $v_k, k = 0, 1, 2$, we first construct the rest frame boost of $B(v_0)$. Since $B(u_0) = B(v_0)$, the boosts in polar decomposition and Minkowski Gram-Schmidt are equal, $B^{\mathrm{PD}} = B^{\mathrm{GS}}$. To prove that the rotation parts agree as well, i.e. that $R^{\mathrm{PD}} = R^{\mathrm{GS}}$, we first show that the set of $\mathrm{GS}^3$ vectors obtained from the spatial parts of $B(v_0)v_1$ and $B(v_0)v_2$, embedded in a set of four-vectors with zero energy component, is equal to the set given by the last three orthonormal four-vectors obtained from $\mathrm{GS}^4$ of $B(v_0)v_0$, $B(v_0)v_1$, and $B(v_0)v_2$, i.e. $\{(0, \mathrm{GS}^3(\overrightarrow{B(v_0)v_1}, \overrightarrow{B(v_0)v_2})_k)\}_{k=0,1,2} = \{\mathrm{GS}^4(B(v_0)v_0, B(v_0)v_1, B(v_0)v_2)_k\}_{k=1,2,3}$. Indeed,

$$u_0 = \mathrm{norm}_4(B(v_0)v_0) = (1, \vec{0}) \ ,$$

$$u_1 = \mathrm{norm}_4(v_1 - (v_1^0, \vec{0})) = (0, \mathrm{norm}_3(v_1))$$

$$u_2 = \mathrm{norm}_4(v_2 - (v_2^0, \vec{0})) - (0, \vec{u}_1(\vec{v}_2 \cdot \vec{u}_1)) = \mathrm{norm}_4((0, \vec{v}_2) - (0, \vec{u}_1(\vec{v}_2 \cdot \vec{u}_1)))$$
$$= (0, \mathrm{norm}_3(\vec{v}_2 - \vec{u}_1(\vec{v}_2 \cdot \vec{u}_1)))$$

$$u_3^\mu = \sum_{\nu,\rho,\sigma,\kappa} g^{\mu\nu} \epsilon_{\nu\rho\sigma\kappa} u_0^\rho u_1^\sigma u_2^\kappa = \sum_{\nu,i,j} g^{\mu k} \epsilon_{k0ij} \underbrace{u_0^0}_{=1} u_1^i u_2^j = \sum_{k,i,j} g^{\mu k} \epsilon_{k0ij} u_1^i u_2^j$$

$$= \begin{cases} 0 \text{ if } \mu = 0 \\ \sum_{k,i,j} \epsilon_{0kij} u_1^i u_2^j = [\vec{u}_1 \times \vec{u}_2]_k \text{ if } \mu = k \end{cases}. \tag{22}$$

Note that $\langle u_0, u_0 \rangle = 1$ and $\langle u_i, u_i \rangle = -1$ for $i = 1, 2, 3$. Therefore, the first component of the $\mathrm{GS}^4$ vectors other than $u_0$ is always zero while the spatial components are equal to those obtained from $\mathrm{GS}^3$ (cf. Eq. (25) below). Since $\mathrm{GS}^4$ is Lorentz-equivariant, we have $\mathrm{GS}^4(B(v_0)v_0, B(v_0)v_1, B(v_0)v_2) = B(v_0)\mathrm{GS}^4(v_0, v_1, v_2) = B(v_0)\{u_k\}_{k=0,1,2,3}$. Taken together, we have thus shown that the pure rotation $R^{\mathrm{PD}}$ is the Lorentz transformation which in its rows contains the covectors corresponding to the orthonormal four-vectors $\{B(v_0)u_k\}_{k=0,1,2,3}$, namely:

$$R^{\mathrm{PD}} = \begin{pmatrix} \text{---} \ u_0^T B(v_0)^T g \ \text{---} \\ \text{---} \ u_1^T B(v_0)^T g \ \text{---} \\ \text{---} \ u_2^T B(v_0)^T g \ \text{---} \\ \text{---} \ u_3^T B(v_0)^T g \ \text{---} \end{pmatrix} = \begin{pmatrix} \text{---} \ u_0^T g B(v_0)^{-1} \ \text{---} \\ \text{---} \ u_1^T g B(v_0)^{-1} \ \text{---} \\ \text{---} \ u_2^T g B(v_0)^{-1} \ \text{---} \\ \text{---} \ u_3^T g B(v_0)^{-1} \ \text{---} \end{pmatrix} = R^{\mathrm{GS}}, \tag{23}$$

where in the last step we have inserted $I_4 = gg$ and then identified $B(v_0)^{-1} = gB(v_0)^T g$. We indeed find that also the rotational part of the two local reference frames is the same which concludes the proof.

Since $L^{\mathrm{PD}} = L^{\mathrm{GS}}$, the correct transformation behavior $L \to L\Lambda^{-1}$ of the local reference frame based on Alg. 1 immediately follows from the correct transformation behavior of $L^{\mathrm{GS}}$.

## D   Experimental details

### D.1   Constructing local reference frames

Here we give additional details on the local reference frame construction presented in Sec. 4.5. We use double precision for precision-critical operations to minimize sources of equivariance violation. In particular, we use double precision for all operations in the vector prediction (13) and the polar decomposition in Algorithm 1, except for the neural network $\varphi_k(s_i, s_j, \langle p_i, p_j \rangle)$ which is evaluated in single precision.

**Equivariant vector prediction.** We use the same architecture for the network $\varphi_k$ in all experiments, an MLP with 2 layers and 128 hidden channels. We find that normalizing the result of Eq. (13) improves numerical stability. The modified relation reads

$$v'_{i,k} = v_{i,k} \Big/ \sqrt{\sum_{i=1}^{n} \|v_{i,k}\|^2},\tag{24}$$

where $\|a\| = \sqrt{\langle a, a \rangle}$ and $i = 1 \ldots N$ runs over the $N$ particles in the set. Note that $\| v_{i,k} \| \geq 0$ and $\|p_i + p_j\| \geq 0$ by construction.

**3D Gram-Schmidt orthonormalization.** Here we describe the operation $\mathrm{GramSchmidt}(\vec{w}_1, \vec{w}_2)$ (or $\mathrm{GS}^3(\vec{w}_1, \vec{w}_2)$) in Algorithm 1 in more detail

$$\begin{aligned}
\mathrm{norm}_3(\vec{w}) &:= \frac{\vec{w}}{\|\vec{w}\| + \epsilon}, \\
\vec{u}_1 &= \mathrm{norm}_3\left(\vec{w}_1\right), \\
\vec{u}_2 &= \mathrm{norm}_3\left(\vec{w}_2 - \vec{u}_1(\vec{w}_2 \cdot \vec{u}_1)\right), \\
\vec{u}_3 &= \vec{u}_1 \times \vec{u}_2.
\end{aligned}\tag{25}$$

We use $\epsilon = 10^{-15}$, and $\times$ denotes the cross product. The resulting vectors $\vec{u}_1, \vec{u}_2, \vec{u}_3$ are orthonormal, i.e. they satisfy $\vec{u}_i \cdot \vec{u}_j = \delta_{ij}$.

**Numerical stability of local frame prediction.** Most particles at the LHC can be considered massless as the typical energy scale is much larger than the mass of single particles. When operating on four-momenta using single precision, the particle mass value is often modified due to numerical underflows, leading to particles with zero mass in some cases. To avoid downstream instabilities, we enforce a minimal particle mass $m_\epsilon$ by increasing the energy of all input particles $p_i$ as $E' = \sqrt{m_\epsilon^2 + E^2}$, hence $m'^2 = m_\epsilon^2 + m^2$. We use $m_\epsilon = 10^{-5}$ and $m_\epsilon = 5 \cdot 10^{-3}$ for the amplitude regression and tagging experiments, respectively.

Furthermore, we have carefully designed Eq. (13) to avoid vectors $v_{i,0}$ with small or negative norm which would cause instabilities due to large boosts. The reason that local frames with large boosts $B$ are problematic is that they multiply the latent features in each message passing step, possibly leading to exploding gradients. Such large boosts are caused by vectors $v_{i,0}$ constructed in Eq. (13) that have small but positive norm, because they yield large values for $\gamma$ in Eq. (2). These small-norm vectors $v_{i,0}$ can be caused by the typically small-norm particle 4-momenta $p_i$ in Eq. (13) either if all 4-momenta $p_i$ point in a similar direction, or if the coefficient $\mathrm{softmax}(\varphi_k(\ldots))$ is small for all except one particle.

Even bigger problems could occur if the vectors $v_{i,0}$ had zero or negative norm, because the boost matrix $B$ in Eq. (2) is ill-defined due to $\vec{\beta}^2 \geq 1$. In practice, such pathological vectors $v_{i,0}$ do not occur with our implementation.

Moreover, we have identified several techniques to avoid large boosts already at initialization, where such instabilities are most likely to occur. First, we include a softmax operation in Eq. (13) to prevent any negative-norm vectors in the process. Second, we use regulator masses to enforce a lower limit on the norms of the input particles $p_i$. Third, we multiply by $p_i + p_j$ instead of simply $p_i$ in Eq. (13), because $p_i + p_j$ typically is not strongly boosted anymore.

Lastly, the $\mathrm{SO}(3)$ Gram-Schmidt orthonormalization used in Alg. 1 requires two linearly independent vectors. We ensure that the two vectors $\vec{w}_1, \vec{w}_2$ are not collinear nor null by calculating the cross-product between the two vectors. Concretely, if $\|\vec{w}_1 \times \vec{w}_2\| < \epsilon_{\mathrm{collinear}}$, we replace $\vec{w}_1$ and $\vec{w}_2$ by $\vec{w}_1 + \epsilon_{\mathrm{collinear}} \vec{\delta}_1$ and $\vec{w}_2 + \epsilon_{\mathrm{collinear}} \vec{\delta}_2$, respectively, where $\vec{\delta}_{1,2}$ are random normal directions, i.e. $\delta_i \sim \mathcal{N}(0, 1)$. In our experiments, we set $\epsilon_{\mathrm{collinear}} = 10^{-16}$ and we observe that, besides a handful of exceptions at initialization, the predicted vectors are never regularized. This indicates that our orthonormalization procedure is well under control.

## D.2 QFT amplitude regression

**Dataset.** The amplitude regression datasets are publicly available on https://zenodo.org/records/16793011. They are generated in complete analogy to Spinner et al. [42]. The Monte

Carlo generator MadGraph [1] is used to generate particle configurations for a given process following the phase-space distribution predicted in QFT. Each data point contains the kinematics of the two colliding quarks, the intermediate $Z$ bosons, and the additional gluons. The events have unit weight and, therefore, closely simulate the distributions which can be observed in the experiments. The amplitude is re-evaluated using a standalone MadGraph package and stored with the particle kinematics. Physics-motivated cuts are needed to avoid divergent regions. We apply a cut on the transverse momentum, $p_T > 20$ GeV, and on the angular distance between gluons, $\Delta R > 0.4$.

The $Z + \{1, 2, 3\}g$ datasets contain 10M events each, while we generate 100M events for the more challenging $Z + 4g$ dataset to complete our scaling studies. For validation and testing, we use the same independent dataset as in [42], from which we use 100k events for validation and 500k events for the final evaluation. The amplitudes $A$ are preprocessed with a logarithmic transformation followed by a standardization,

$$A' = \frac{\log A - \overline{\log A}}{\sigma_{\log A}} \ . \tag{26}$$

The input four-momenta are rescaled by the standard deviation of the entire dataset. Finally, we remove the alignment along the z-axis in two steps. First, we boost the inputs in the reference frame of the sum of the two incoming particles, also called center-of-mass frame. Then, we apply a random general Lorentz transformation.

**Models.** Our GNN and LLoCa-GNN use standardized four-momenta and one-hot encoded particle types as inputs. For LLoCa-GNN, the four-momenta are transformed into their local frames. Both networks share the same graph neural network backbone. The GNN consists of three edge convolution blocks. In each convolution, the message passing network is an MLP with three hidden layers with 128 hidden channels. The message between nodes is constructed from the hidden node representations and one additional edge feature defined as the Minkowski product of the two particle's four-momenta. The aggregation of the messages is done with a simple summation. For the LLoCa-GNN, we use 64 channels each for scalar and vector representations, i.e. 64 scalars and 16 vectors. The GNN and the LLoCa-GNN have $2.5 \cdot 10^5$ parameters and $2.7 \cdot 10^5$ parameters, respectively.

Our Transformer and LLoCa-Transformer networks use the same inputs as the GNN and LLoCa-GNN described above. These features are encoded with a linear layer in a latent representation with 128 channels. Each transformer block performs a multi-head self-attention with eight heads followed by a fully-connected MLP with two layers and GELU nonlinearities. Similar to the LLoCa-GNN, the latent representation of the LLoCa-Transformer is divided into 50% scalar and 50% vector features. We stack eight transformer blocks, totaling $10^6$ parameters for both networks. The LLoCa-Transformer has $2 \times 10^4$ additional parameters.

The other Lorentz-equivariant baseline, L-GATr, is taken from its official repository[5]. We only modify the number of hidden multivector channels to 20, such that the number of parameters of all our transformer models is roughly $10^6$, matching the LLoCa-Transformer. The inputs of the MLP-I baseline are all the possible Lorentz-invariant features. The neural networks consists of five hidden layers with 128 hidden channels each, summing up to $5.5 \times 10^4$ learnable parameters.

**Training.** All models are trained with a batch size of 1024 and using the Adam optimizer with $\beta = [0.99, 0.999]$ to optimize the network weights for $2 \cdot 10^5$ iterations. We use PyTorchs `ReduceLROnPlateau` learning rate scheduler to decrease the learning rate by a factor 0.3 if no improvements in the validation loss are observed in the last 20 validations. We validate our models every $10^3$ iterations; if $10^3$ iterations are less than 50 epochs then we validate every 50 epochs. We use the same settings for the L-GATr baseline after finding improved accuracy over the original L-GATr training parameters. The models are trained on a single A100 GPU.

**Data augmentation.** In our framework, to train with data augmentation we have to sample global reference frames that are elements of the respected symmetry group. We follow the polar decomposition discussed in App. C to define global frames. We sample uniformly distributed rotations represented as quaternions. While it is feasible to uniformly sample rotation matrices on the 3-sphere, the non-compactness of the Lorentz group makes it impossible to uniformly sample boost matrices. We therefore sample each component of the boost velocity from a Gaussian distribution $\mathcal{N}(0.0, 0.1)$,

---

[5]https://github.com/heidelberg-hepml/lgatr

truncated at three standard deviations. The mean and standard deviation of the Gaussian is estimated based on the training dataset. Finally, we define the global reference frames as the combination of a boost to the center-of-mass frame of the two incoming particles and the random Lorentz transformation described above.

**Timing and FLOPs.** We evaluate the total training time and the FLOPs per forward pass for the amplitude regression networks. Training times are measured within our code environment on an H100 GPU, excluding overhead from validation and testing. FLOPs per forward pass are computed for $Z + 4g$ events using PyTorch's `torch.utils.flop_counter.FlopCounterMode`. The results for both training time and FLOPs are summarized in Fig. 3.

### D.3 Jet tagging

**Dataset.** We use the JetClass tagging dataset of Qu et al. [38].[6] The data samples are organized as point clouds, simulated in the CMS experiment environment at detector level. For details on the simulation process, see Qu et al. [38]. The number of particles per jet varies between 1 and 128, typically averaging 30–50 depending on the jet type. Each particle is described by its four-momentum, six features encoding particle type, and four additional variables capturing trajectory displacement. The dataset comprises 10 equally represented classes and is split into 100 million events for training, 20 million for testing, and 5 million for validation. Accuracy and AUC are used as global evaluation metrics, and class-specific background rejection rates at fixed signal efficiency are reported in Tab. 5.

**Models.** Our implementations of LLoCa-ParticleNet and LLoCa-ParT are based on the latest official versions of ParticleNet and ParT.[7] We apply minimal modifications to the original codebases. In LLoCa-ParticleNet, we transform the sender's local features from its own local frame to that of the receiver, and perform the k-nearest-neighbors search using four-momenta expressed in each node's local frame. In LLoCa-ParT, we similarly evaluate edge features – used to compute the attention bias – based on four-momenta in local frames. Additionally, we implement tensorial message passing in the particle attention blocks, as described in Sec. 4.3, while retaining standard scalar message passing for class attention. Unlike the official ParT implementation, which uses automatic mixed precision, our LLoCa-ParT implementation employs single precision throughout to avoid potential equivariance violations caused by numerical precision limitations.

Our baseline transformer is a simplified variant of ParT, maintaining a similar overall network architecture. Whereas ParT aggregates particle information using class attention, our baseline instead employs mean aggregation. Furthermore, the learnable attention bias present in ParT is omitted in our design, leading to a significant gain in evaluation time. Consistent with ParT, we use 128 hidden channels and expand the number of hidden units in each attention block's MLP by a factor of four. While ParT includes eight particle attention blocks followed by two class attention blocks, our baseline transformer consists of ten standard attention blocks, followed by mean aggregation.

The inputs to all our models are the particle four-momenta $p_i$ combined with the 17 particle-wise features described in Tab. 3. Following Eq. (13), the Frames-Net $\varphi$ receives pairwise inner products based on the four-momenta $p_i$ combined with the 17 particle-wise features in Tab. 3 as scalars $s_i$. Note that the 7 kinematics features are not invariant under Lorentz transformations, we discuss this aspect below in more detail. For the backbone network, the kinematic features $\Delta\eta, \Delta\phi, \log p_T, \log E, \log p_T/p_{T,\mathrm{jet}}, \log E/E_{\mathrm{jet}}, \Delta R$ are all expressed in the respective local reference frames. The 6 particle identification features and the 4 trajectory displacement features are treated like scalars, i.e. they are not transformed when moving to the local reference frames. The backbone network input is then constructed from these combined 17 features.

For all models, we break Lorentz equivariance by including additional symmetry-breaking inputs in two ways. First, the 7 kinematic features listed in Tab. 3 are only invariant under the unbroken $SO(2)$ subgroup of rotations around the beam axis, but included in the list of Lorentz scalars $s_i$ that serves as input to the Frames-Net. Second, we append additional three "reference particles" to the particle set that serves as input for the local reference frame prediction described in Sec. 4.5. These reference particles are the time direction $(1, 0, 0, 0)$ as well as the beam direction and its counterpart $(1, 0, 0, \pm 1)$. Together, they define the time and beam directions as two reference directions which will

---

[6]Available at https://zenodo.org/records/6619768 under a CC-BY 4.0 license.
[7]Available at https://github.com/hqucms/weaver-core/tree/dev/custom_train_eval.

| Category | Variable | Definition |
|---|---|---|
| Kinematics | $\Delta\eta$ | difference in pseudorapidity $\eta$ between the particle and the jet axis |
| | $\Delta\phi$ | difference in azimuthal angle $\phi$ between the particle and the jet axis |
| | $\log p_T$ | logarithm of the particle's transverse momentum $p_T$ |
| | $\log E$ | logarithm of the particle's energy |
| | $\log \frac{p_T}{p_{T(\text{jet})}}$ | logarithm of the particle's $p_T$ relative to the jet $p_T$ |
| | $\log \frac{E}{E(\text{jet})}$ | logarithm of the particle's energy relative to the jet energy |
| | $\Delta R$ | angular separation between the particle and the jet axis $(\sqrt{(\Delta\eta)^2 + (\Delta\phi)^2})$ |
| Particle identification | charge | electric charge of the particle |
| | Electron | if the particle is an electron (\|pid\|=11) |
| | Muon | if the particle is a muon (\|pid\|=13) |
| | Photon | if the particle is a photon (pid=22) |
| | CH | if the particle is a charged hadron (\|pid\|=211 or 321 or 2212) |
| | NH | if the particle is a neutral hadron (\|pid\|=130 or 2112 or 0) |
| Trajectory displacement | $\tanh d_0$ | hyperbolic tangent of the transverse impact parameter value |
| | $\tanh d_z$ | hyperbolic tangent of the longitudinal impact parameter value |
| | $\sigma_{d_0}$ | error of the measured transverse impact parameter |
| | $\sigma_{d_z}$ | error of the measured longitudinal impact parameter |

Table 3: **Input features for the JetClass dataset [38]**. Kinetic features in the global frame are used as symmetry breaking during prediction of the local frames.

stay fixed for all inputs, effectively breaking the Lorentz-equivariance down to the transformations that keep these two directions unchanged. We note that two of the three reference vectors would already be sufficient, and also already one of the two ways of Lorentz symmetry breaking would be sufficient in principle. The effect of symmetry-breaking inputs is discussed in App. E.

**Training.** For both LLoCa-ParT and LLoCa-ParticleNet, we adopt the same training hyperparameters as the official implementations, without any additional tuning. Specifically, models are trained for 1,000,000 iterations with a batch size of 512, using the Ranger optimizer as implemented in the official ParT and ParticleNet repositories. A constant learning rate is applied for the first 700,000 iterations, followed by exponential decay. We use a learning rate of 0.001 for all networks.

Our Transformer and LLoCa-Transformer models are trained with the same number of iterations, batch size, and initial learning rate as LLoCa-ParT, but use the AdamW optimizer in conjunction with a cosine annealing learning rate schedule.

**Timing and FLOPs.** To compare the training costs of the different tagging networks, we evaluate both training time and FLOPs per forward pass. Training times are measured within our code environment on an H100 GPU, excluding overhead from data loading, validation, and testing. FLOPs per forward pass are computed for events containing 50 particles using PyTorch's `torch.utils.flop_counter.FlopCounterMode`. The results for both training time and FLOPs are summarized in Tab. 1.

# E  Additional results

In this section, we present additional results for the QFT amplitude regression and jet tagging experiments.

## E.1  QFT amplitude regression

**More Lorentz-equivariant architectures.** Following the guiding principle of our contribution, we present other architectures that are turned into Lorentz-equivariant via local canonicalization. We study the same graph network without including the Lorentz-invariant edge features $\langle p_i, p_j \rangle$ as arguments for the main network. The other baseline we consider is a simple MLP with four-

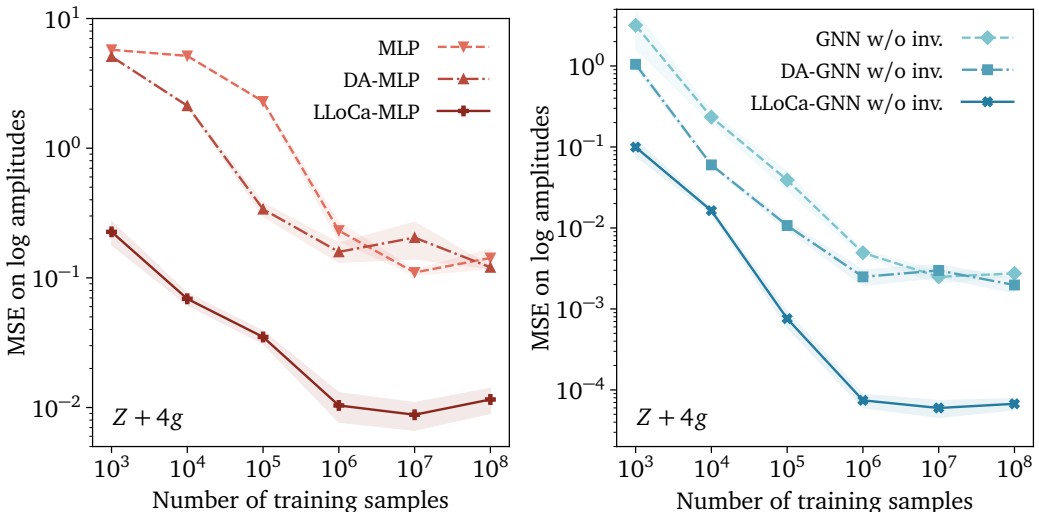

Figure 5: **Comparison between LLoCa models and their non-equivariant counterparts.** We consider a simple MLP (left) and a graph network that does not use additional Lorentz invariant edge features (right). The global frames for the data augmentation (DA) results are sampled as described in App. D.

momenta as inputs. We do not include non-trivial frame-to-frame transformations as in Eq. (10), effectively allowing only scalar messages. The architecture is still Lorentz-equivariant due to Sec. 4.2. In Fig. 5 we repeat the same scaling study as presented in Sec. 5. Although these two models are not competitive with the models presented in Sec. 5, we observe significant improvements from introducing Lorentz-equivariance.

**Frames-Net regularization.** Besides the message representations discussed in Tab. 2, the main design choice of LLoCa is the architecture of the Frames-Net $\varphi$ described in Sec. 4.5. We consider several choices and train LLoCa-Transformers on the $Z + 4g$ dataset for different training dataset sizes, see Fig. 6. First, we find that the default Frames-Net ("Learnable LLoCa") tends to overfit at larger training dataset sizes than the main network. We tackle this by using a dropout rate of 0.2 when less than $10^5$ training events are available ("Learnable LLoCa + Dropout"). We do not use dropout for larger training dataset sizes because we find that it affects the network performance. A similar effect can be achieved by decreasing the size of the Frames-Net ("Learnable small LLoCa"), where we use 16 instead of 128 hidden channels. However, we find that decreasing the network size slightly degrades performance in the large-data regime. We find that a Frames-Net with parameters fixed after initialization ("Fixed LLoCa") still outperforms the non-equivariant counterpart. When little training data is available, a fixed Frames-Net combined with a dropout rate of 0.2 ("Fixed LLoCa + Dropout") even achieves the best performance.

**Constructing local reference frames.** Our default approach for constructing local reference frames based on a polar decomposition is equivalent to a direct Gram-Schmidt algorithm in Minkowski space and removing the boost yields the special case of local reference frames for SO(3)-equivariant architectures, see Sec. 4.5 and App. C. We now put these observations to the test and train LLoCa-Transformers on different training dataset sizes of the $Z + 4g$ dataset, see Fig. 6. First, we find that a direct Gram-Schmidt algorithm in Minkowski space ($GS^4$) shows very similar performance to our default polar decomposition approach (PD). The special case of SO(3)-equivariance obtained by only including the rotation part ($GS^3$) yields significantly worse performance due to the reduced symmetry group. All approaches show a very similar scaling behaviour with the training dataset size.

**Attention inner product.** For the scaled dot-product attention discussed in Eq. (11), we use the Minkowski product to project keys onto queries. We find that this design choice significantly outperforms the naive choice of Euclidean attention, see Tab. 4.

## E.2 Jet tagging

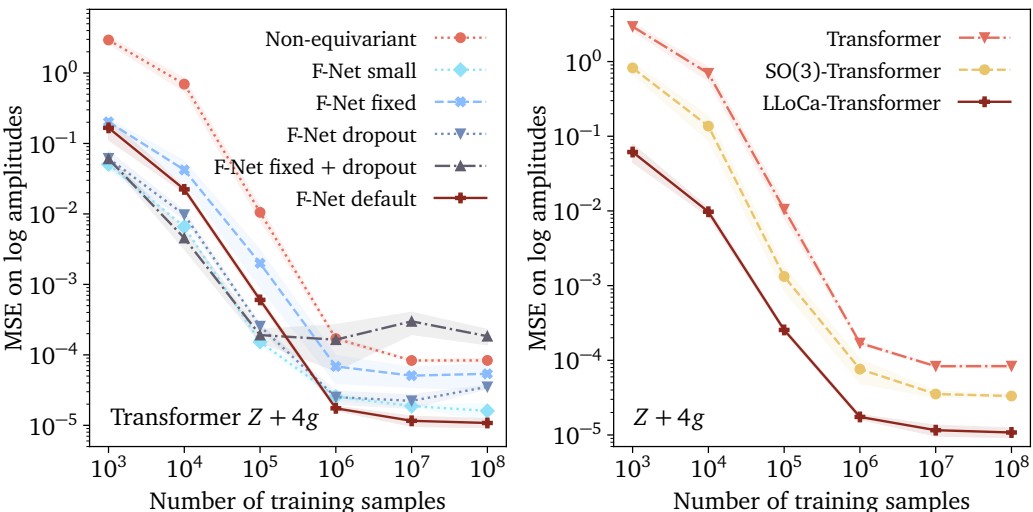

Figure 6: **Design choices when constructing local reference frames.** Left: Effect of the network architecture used for the Frames-Net $\varphi$. We use "Learnable LLoCa" in all other experiments, and "Learnable LLoCa + Dropout" when less than $10^5$ training events are available. Right: Comparison of different orthonormalization schemes. We use "PD" in all other experiments.

| | $H \to b\bar{b}$ Rej$_{50\%}$ | $H \to c\bar{c}$ Rej$_{50\%}$ | $H \to gg$ Rej$_{50\%}$ | $H \to 4q$ Rej$_{50\%}$ | $H \to l\nu q\bar{q}'$ Rej$_{99\%}$ | $t \to bq\bar{q}'$ Rej$_{50\%}$ | $t \to bl\nu$ Rej$_{99.5\%}$ | $W \to q\bar{q}'$ Rej$_{50\%}$ | $Z \to q\bar{q}$ Rej$_{50\%}$ |
|---|---|---|---|---|---|---|---|---|---|
| PFN [26] | 2924 | 841 | 75 | 198 | 265 | 797 | 721 | 189 | 159 |
| P-CNN [41] | 4890 | 1276 | 88 | 474 | 947 | 2907 | 2304 | 241 | 204 |
| ParticleNet [37] | 7634 | 2475 | 104 | 954 | 3339 | 10526 | 11173 | 347 | 283 |
| ParT [38] | 10638 | 4149 | 123 | 1864 | 5479 | 32787 | 15873 | 543 | 402 |
| MIParT-L [48] | 10753 | 4202 | 123 | 1927 | 5450 | 31250 | 16807 | 542 | 402 |
| L-GATr [9] | 12987 | 4819 | 128 | 2311 | 6116 | 47619 | 20408 | 588 | 432 |
| Transformer (ours) | 10753 | 3333 | 116 | 1369 | 4630 | 24390 | 17857 | 415 | 334 |
| LLoCa-Transf.* (ours) | 11628 | 4651 | 125 | 2037 | 5618 | 39216 | 17241 | 548 | 410 |
| LLoCa-ParT* (ours) | 11561 | 4640 | 125 | 2037 | 5900 | 41667 | 19231 | 552 | 419 |
| LLoCa-ParticleNet* (ours) | 7463 | 2833 | 105 | 1072 | 3155 | 10753 | 9302 | 403 | 306 |

Table 5: **Background rejection rates** $1/\epsilon_B$ **for the JetClass dataset [38].** For each class, Rej$_{N\%}$ represents the inverse signal fraction at fixed background rejection rate $N\%$. Lorentz-equivariant methods are marked with an asterisk*. See Tab. 1 for the global accuracy and AUC.

**More evaluation metrics.** In addition to the global evaluation metrics reported in Tab. 1, we report class-wise background rejection rates in Tab. 5.

**Effect of Lorentz symmetry breaking.** We follow the approach of Ref. [9, 42] of using fully Lorentz-equivariant models and breaking their symmetry instead of considering models that are only equivariant under the $SO(2)$ symmetry of rotations around the beam direction. Note that the baseline models ParticleNet, ParT, MIParT and our Transformer are all $SO(2)$-invariant, because their inputs are invariant under $SO(2)$ transformations.

Table 4: **Benefit of Minkowski attention compared to Euclidean attention.** We show results for the LLoCa-Transformer on the full $Z + 4g$ dataset.

| Attention | MSE ($\times 10^{-5}$) |
|---|---|
| Euclidean | $2.5 \pm 0.4$ |
| Minkowski | $1.0 \pm 0.1$ |

As described in App. D, we use two independent methods to break the Lorentz symmetry down to the $SO(2)$ subgroup of rotations around the beam direction. They are (a) including features in the list of Lorentz-invariant inputs $s_i$ in Eq. (13) that are only invariant under the unbroken subgroup (non-invariant scalars, or NIS), and (b) including reference vectors (RV) pointing in the directions orthogonal to the unbroken subgroup as additional particles. The additional NIS and RV inputs are described in more detail in App. D. In Tab. 6 we study the effect of this design choice.

To study the impact of Lorentz symmetry breaking, we directly compare the cases of "No symmetry breaking" with only "RV", only "NIS", and our default of including both "RV & NIS". The Lorentz-equivariant model that does not include any source of symmetry breaking is significantly worse and

| Symmetry breaking | All classes Accuracy | AUC | $H \to b\bar{b}$ $\mathrm{Rej}_{50\%}$ | $H \to c\bar{c}$ $\mathrm{Rej}_{50\%}$ | $H \to gg$ $\mathrm{Rej}_{50\%}$ | $H \to 4q$ $\mathrm{Rej}_{50\%}$ | $H \to l\nu q\bar{q}'$ $\mathrm{Rej}_{99\%}$ | $t \to bq\bar{q}'$ $\mathrm{Rej}_{50\%}$ | $t \to bl\nu$ $\mathrm{Rej}_{99.5\%}$ | $W \to q\bar{q}'$ $\mathrm{Rej}_{50\%}$ | $Z \to q\bar{q}$ $\mathrm{Rej}_{50\%}$ |
|---|---|---|---|---|---|---|---|---|---|---|---|
| Non-equivariant | 0.855 | 0.9867 | 10753 | 3333 | 116 | 1369 | 4630 | 24390 | 17857 | 415 | 334 |
| SO(3)-equi. RV & NIS | 0.863 | 0.9880 | 11976 | 4619 | 124 | 2030 | 5571 | 37037 | 16394 | 545 | 407 |
| No symmetry breaking | 0.856 | 0.9870 | 9756 | 3781 | 113 | 1660 | 4762 | 24692 | 15385 | 465 | 351 |
| RV | 0.861 | 0.9877 | 11494 | 4444 | 122 | 1923 | 5222 | 34483 | 17391 | 521 | 398 |
| NIS | 0.863 | 0.9881 | 11300 | 4630 | 124 | 1983 | 5249 | 40817 | 17544 | 547 | 405 |
| RV & NIS (default) | 0.864 | 0.9882 | 11628 | 4651 | 125 | 2037 | 5618 | 39216 | 17241 | 548 | 410 |

Table 6: **Impact of symmetry breaking for jet tagging.** LLoCa-Transformer accuracy, AUC, and background rejection $1/\epsilon_B$ for different symmetry-breaking approaches. We compare the effect of symmetry breaking with reference vectors (RV) and non-invariant scalars (NIS) with the case of no symmetry breaking as well as a non-equivariant and a SO(3)-equivariant model (which also uses symmetry breaking with RV&NIS).

Table 7: LLoCa significantly lifts the performance of non-equivariant backbone architectures even when using the same computational resources. See Tab. 1 for more baselines.

| Network | Training time | Accuracy | AUC |
|---|---|---|---|
| Transformer | 15 h | 0.855 | 0.9867 |
| LLoCa-Transformer ($0.5\times$ iterations) | 15 h | 0.861 | 0.9877 |
| Transformer ($2\times$ iterations) | 31 h | 0.855 | 0.9868 |
| L-GATr ($0.2\times$ iterations) | 33 h | 0.855 | 0.9869 |
| ParT | 33 h | 0.861 | 0.9877 |
| LLoCa-Transformer | 31 h | 0.864 | 0.9882 |

only marginally better than the non-equivariant approach, because it constrains the model to assign an equal tagging score to input jets with different physical meaning. The models trained with symmetry breaking through the "RV", "NIS" and "RV & NIS" approaches all have a sufficient amount of symmetry breaking included, but still yield slightly different results. We also compare to a SO(3)-equivariant model using the same reference vectors and non-invariant scalars to break the symmetry down to the unbroken subgroup of rotations around the beam direction.

**Efficiency of LLoCa taggers at equal computational cost.** As demonstrated in Tab. 1, LLoCa taggers consistently improve the performance of backbone networks such as vanilla transformers, at the cost of increased computational cost. In Tab. 7, we present additional results for networks trained with equal computational cost. The main baselines are ParT [38], widely used by experimental collaborations, and the recently proposed L-GATr [42], which improves upon ParT but requires $4\times$ longer training (166h). Our most compute-efficient LLoCa tagger is the LLoCa-Transformer. It almost matches L-GATr's accuracy yet trains in only 31h (vs 33h for ParT), delivering extra performance at lower cost – one of the key results of this paper.

We also matched GPU hours by training the plain transformer for $2\times$ iterations and the LLoCa-Transformer for $0.5\times$ iterations. Again, we find that the LLoCa-Transformer consistently achieves improvements at fixed training time (see Tab. 7).

