# OpenReview forum: "Lorentz Local Canonicalization: How to make any Network Lorentz-Equivariant"
_NeurIPS.cc/2025/Conference — NeurIPS 2025 poster_

### Official Review · Reviewer_VHeN · 2025-07-01

**Clarity:** 4
**Significance:** 3
**Originality:** 2
**Rating:** 4
**Confidence:** 5

**Summary:**

The paper introduces **Lorentz Local Canonicalization (LLoCa)**, a general framework that converts any backbone, e.g., graph network, transformer, or MLP, into an *exactly* Lorentz-equivariant model without bespoke equivariant layers. The authors use a simple Lorentz-equivariant architecture to predict three four-vectors for each particle. The four-vectors are then used to construct the local reference frames following a deterministic orthogonalization algorithm. The authors extend recent tensorial message-passing ideas to the non-compact Lorentz group and derive a Minkowski dot-product attention that reuses efficient vanilla attention code. The approach incurs only a moderate computational overhead, with a 10–30% increase in FLOPs and a 30–110% increase in training time compared to non-equivariant baselines. Compared to other SOTA Lorentz-equivariant architectures, the models train up to 4x faster and use 5–100x fewer FLOPs.

**Questions:**

1. [2] prove that, for a certain domain, no *continuous* canonicalization map exists for the compact orthogonal group. Is the vierbein predicted by LLoCa a continuous function of the input four-vectors and is the overall architecture of LLoCa continuous w.r.t. the input?
2. Can the canonicalization procedure be extended from $O(1,3)$ to the general family $O(1,d)$ as considered in [3]? It is recommended for a brief discussion over this generalization.

**Ethical Concerns:**

["NO or VERY MINOR ethics concerns only"]

**Final Justification:**

The rebuttal addressed all of my questions. The work is clear and the empirical result is strong, though its theoretical novelty is moderate.

**Limitations:**

The authors have discussed the limitations in line 226 and 317. It is recommended for the authors to write them into one section and give possible future solutions for them.

**Paper Formatting Concerns:**

I do not have paper formatting concerns and the authors have done a good job over the presentation.

**Quality:**

3

**Strengths And Weaknesses:**

**Strengths**:

1. LLoCa generalizes the canonicalization strategy of [1] from compact groups to the non-compact Lorentz group and can be grafted onto any backbone (graph network, transformer, MLP) without bespoke equivariant layers, widening the design space beyond previous Lorentz-specific networks.
2. Relative to the corresponding non-equivariant model, LLoCa adds only 10–30% FLOPs and 30–110% training time, yet remains 4x faster and 5–100x lighter than L-GATr, the prior best equivariant architecture.



**Weaknesses**:

1. The result accuracy is not the SOTA in the JetClass dataset, contradicting the claim in the abstract. Also, the performance lift over the chosen backbone is marginal, while training time nearly doubles. Results obtained under the equal training time of the backbone and the backbone with the proposed technique would clarify whether the trade-off is warranted.
2. Canonicalization has been comprehensively explored in [1–4] as a special case of frame averaging [5]; especially, Lorentz canonicalization via generalized Gram–Schmidt for Minkowski metric is already discussed in [3], possibly contradicting the first contribution claim the authors make. The paper should state precisely in the main text how LLoCa diverges from, or improves upon, these methods to justify its novelty.

[1] Kaba, Sékou-Oumar, et al. "Equivariance with learned canonicalization functions." International Conference on Machine Learning. PMLR, 2023.

[2] Dym, Nadav, Hannah Lawrence, and Jonathan W. Siegel. "Equivariant frames and the impossibility of continuous canonicalization." arXiv preprint arXiv:2402.16077 (2024).

[3] Lin, Yuchao, et al. "Equivariance via minimal frame averaging for more symmetries and efficiency." arXiv preprint arXiv:2406.07598 (2024).

[4] Ma, George, et al. "A canonicalization perspective on invariant and equivariant learning." arXiv preprint arXiv:2405.18378 (2024).

[5] Puny, Omri, et al. "Frame averaging for invariant and equivariant network design." arXiv preprint arXiv:2110.03336 (2021).

---

> ### Author Rebuttal · Authors · 2025-07-30
>
> Thank you for the thorough and constructive review. We are happy to hear that you appreciate the flexibility and efficiency of our approach. Thanks as well for the questions and criticisms, which we address one by one.
>
> ### 1 SOTA on JetClass dataset
>
> > The result accuracy is not the SOTA on the JetClass dataset
>
> We all agree that given the same training and test data, overall accuracy depends not only on the architecture but also on the many concomitant design choices / hyperparameters ranging from number of channels to choice of optimizer to training schedule. Our focus here was the architecture, and it is for this reason that LLoCa-ParT, Transformer, and LLoCa-Transformer models all use the default ParT settings [7]. This uniform configuration was selected to ensure a fair and consistent comparison between LLoCa architectures and their non-equivariant baselines.
>
> In contrast, the L-GATr results reported in [6] and cited in our original Table 1 rely on carefully tuned hyperparameters. For an apples-to-apples comparison, we have retrained L-GATr with ParT settings and found the following:
>
> Network | Training setup | Accuracy | AUC
> --- | --- | --- | ---
> L-GATr | L-GATr [6] | 0.866 | 0.9885
> L-GATr | ParT [7] (new) | 0.864 | 0.9882
> LLoCa-Transformer | ParT [2] | 0.864 | 0.9882
>
> We are confident that LLoCa-Transformer accuracy could be further increased with hyperparameter tuning, but this was not our primary aim here and also not feasible within the rebuttal period. We will amend the abstract to state that we achieve "competitive and SOTA results".
>
> ### 2 Performance lift from LLoCa
>
> > Also, the performance lift over the chosen backbone is marginal, while training time nearly doubles. Results obtained under the equal training time of the backbone and the backbone with the proposed technique would clarify whether the trade-off is warranted.
>
> Although LLoCa improves accuracy/AUC only in the third decimal place, the gain is nonetheless meaningful. A more tangible metric is the class‑wise background‑rejection rate in Table 5: doubling this rate effectively doubles the usable data when the analysis is background‑limited. For instance, for the $H\to c\bar c$, $H\to 4q$ and $t\to bq\bar q'$ classes LLoCa-Transformer increases the background rejection rates by 40% or more compared to the baseline Transformer. Neural jet taggers are used in almost every LHC analysis, turning already smaller performance gains into real-world money savings because the LHC has to run for less time to collect the same statistical power.
>
> LLoCa adds full Lorentz equivariance to lightweight backbones like a vanilla Transformer. The main baselines are ParT, widely used by experimental collaborations, and the recently proposed L‑GATr, which improves upon ParT but requires $5\times$ longer training (180h). Our LLoCa‑Transformer almost matches L‑GATr’s accuracy yet trains in only 33h (vs. 38h for ParT), delivering extra performance at lower cost - one of the paper’s key results. For LLoCa‑ParT and LLoCa‑ParticleNet, however, we agree that the incremental gain may not justify the additional effort.
>
> We also matched GPU hours by training the plain transformer for $2\times$ iterations and the LLoCa-Transformer for $0.5\times$ iterations. Again, we find that the LLoCa-Transformer consistently achieves improvements at fixed training time.
>
> Network | Training time | Accuracy | AUC
> --- | --- | --- | ---
> Transformer | 17h | 0.855 | 0.9867
> LLoCa-Transformer (0.5x iterations) | 17h | 0.861 | 0.9877
> Transformer (2x iterations) | 34h | 0.855 | 0.9868
> ParT | 38h | 0.861 | 0.9877
> LLoCa-Transformer | 33h | 0.864 | 0.9882
>
> ### 3 Relation of LLoCa to existing canonicalization approaches
>
> > Canonicalization has been comprehensively explored in [1–4] as a special case of frame averaging [5]; especially, Lorentz canonicalization via generalized Gram–Schmidt for Minkowski metric is already discussed in [3], possibly contradicting the first contribution claim the authors make. The paper should state precisely in the main text how LLoCa diverges from, or improves upon, these methods to justify its novelty.
>
> In [1], the authors propose a learned canonicalization function that maps the entire input sample to a single canonical orientation. The key distinction of these 'global canonicalization' methods to our 'local canonicalization' is that LLoCa does not canonicalize the whole sample into one orientation, but instead predicts a unique local coordinate frame for each particle or node in the sample. The methods in [2–5] likewise only study global canonicalization functions. Our LLoCa method is a strict generalization of global canonicalization. In our experiments, we find that local canonicalization outperforms global canonicalization, see Table 2.
>
> Furthermore, the frame‑averaging techniques described in [2,4,5] achieve equivariance by rotating the global input into several canonical poses, passing each through the network, and averaging the resulting outputs. In principle, our method of using just a single frame per particle can be extended to frame-averaging methods that use multiple reference frames. We view this as a direction for future work.
>
> Finally, although [3] implements a canonicalization approach in Minkowski space to construct Lorentz‑equivariant networks, it remains a global technique. To the best of our knowledge, LLoCa is the first framework to realize local canonicalization for exact Lorentz equivariance. Importantly, our experiments demonstrate that local canonicalization without tensorial message passing does not work for the reasons given in Sec. 4.3 and [8], see Tab. 2. Therefore, our generalization of tensorial messages [8] to the Lorentz group provides a key ingredient in making local canonicalization a practically useful method for realizing Lorentz equivariance.
>
> We will expand the related work section of the revised manuscript to include a more in-depth discussion about the difference between local and global canonicalization, and add the possibility of further work in the direction of frame averaging. For clarity, we will remove the brackets around the word "(local)" in the first contribution in the revised manuscript.
>
>
> ### 4 Continuity of the LLoCa framework
>
> > Is the vierbein predicted by LLoCa a continuous function of the input four-vectors and is the overall architecture of LLoCa continuous w.r.t. the input?
>
> In [2] it is proven that in $\mathrm{SO}(d)$ and in the special case $\mathrm{SO}(3)$ it is not possible to find a continuous canonicalization as a function of the input vectors. As $\mathrm{SO}(3)$ is a subgroup of $\mathrm{SO}(1,3)$, our local frame prediction can in general not be a continuous function of the input four-vectors. But in practice, we do not find this to be a problem. For a detailed answer about the numerics of the learned local frames, please take a look at our answer to Reviewer 1.
>
> [2] and [9] argue that with frame averaging of specific weighted frames, the discontinuity of the function can be mitigated. This method can in principle be used in our framework to create a smoother prediction of local frames and is a promising avenue for future research.
>
>
> ### 5 Extension of LLoCa to $\mathrm O(1,d)$
>
> > Can the canonicalization procedure be extended from $\mathrm O(1,3)$ to the general family $\mathrm O(1,d)$ as considered in [3]?
>
> Yes, our method can be extended to the general family $\mathrm O(1,d)$. Note that LLoCa does not cover the full Lorentz group including parity and time reversal, but only the fully-connected subgroup denoted by $\mathrm{SO}^+(1,3)$. Extending LLoCa to $\mathrm O(1,3)$ requires to also predict extra parity and time reversal transformations, see [8] for how this can be done for $\mathrm O(3)$. Extending LLoCa to $\mathrm{SO}^+(1,d)$ is straight-forward. Using Eq. (13), one may predict $d$ many $d+1$-dimensional vectors. Then, a trivial extension of Algorithm 1 with $d$-dimensional Gram-Schmidt algorithm for the spatial rotation can be used to predict orthonormal local frames for canonicalization w.r.t. $\mathrm{SO}^+(1,d)$. Furthermore, the tensorial message passing based on the tensor representations defined by Eq. (3) can be extended directly to the case of $\mathrm O(1,d)$.
>
> ### 6 Limitations and future solutions
>
> > The authors have discussed the limitations in line 226 and 317. It is recommended for the authors to write them into one section and give possible future solutions for them.
>
> Following the suggestion of the reviewer, we will include a section in our revised manuscript that combines the limitations of our framework with suggestions for possible future solutions, similar to the answers provided here in the author response.
>
> We hope we were able to address your questions and look forward to discussing further.
>
> **References**
>
> - [6] J. Spinner et al, "Lorentz-Equivariant Geometric Algebra Transformers for High-Energy Physics", NeurIPS 2024
> - [7] H. Qu et al, "Particle Transformer for Jet Tagging", ICML 2022
> - [8] P. Lippmann et al, "Beyond Canonicalization: How Tensorial Messages Improve Equivariant Message Passing", ICML 2025
> - [9] S. Pozdnyakov and M. Ceriotti, "Smooth, exact rotational symmetrization for deep learning on point clouds", NeurIPS 2023

---

> ### Comment · Reviewer_VHeN · 2025-08-01
> **Official Comment by Reviewer VHeN**
>
> Thank you for the rebuttal; it resolved all my questions, and I have increased my score accordingly.

---

> > ### Author Response · Authors · 2025-08-01
> >
> > Thank you for your response. We are very happy to hear that we were able to address all your questions and we really appreciate your endorsement.

---

### Official Review · Reviewer_79Y8 · 2025-07-01

**Clarity:** 4
**Significance:** 3
**Originality:** 3
**Rating:** 5
**Confidence:** 3

**Summary:**

This paper introduces LLoCa, which can enable Lorentz-equivariant adaptation of arbitrary backbone networks (e.g., Transformers). By predicting local reference frames to transform input data into Lorentz-invariant features, the framework allows backbone networks to process these features and produce equivariant outputs. It addresses the computational inefficiency and architectural inflexibility of existing methods. It achieves state-of-the-art (SOTA) performance in high-energy physics tasks such as Jet tagging and QFT amplitude regression, with significantly improved computational efficiency.

**Questions:**

(1)	For two-particle systems, could virtual particles or preprocessing be introduced?
(2)	Can the framework be modified to support computations for other non-compact groups (e.g., the Poincaré group)?

**Ethical Concerns:**

["NO or VERY MINOR ethics concerns only"]

**Final Justification:**

My concerns about this work are largely alleviated. I will keep my score 5 unchanged.

**Limitations:**

Yes (Section 4.5 explicitly states the particle-count constraint)

**Paper Formatting Concerns:**

Formatting is compliant.

**Quality:**

4

**Strengths And Weaknesses:**

Quality：
1) Rigorous methodology: Polar decomposition for Li (Algorithm 1) achieves strict equivariance, backbone-agnostic compatibility, resolving computational challenges of non-compact groups.
2) Comprehensive experimental design for high-energy physics, including 10-class Jet tagging and QFT amplitude regression, validated on 125M particle jet data.

Clarity：
Theoretical derivations are precise: Eqs.1–12 formally define Lorentz group representations, canonicalization, and message passing. Ablation studies reveal the necessity of mixed tensor representations.

Significance:
The framework solves a critical bottleneck in HEP machine learning: it enables Lorentz-equivariant adaptation of general-purpose backbone networks, providing a tool for analyzing the massive datasets from the Large Hadron Collider (LHC).

Originality:
Novel contributions: Local canonicalization unifies data augmentation and exact equivariance and introduces an innovative Minkowski inner-product attention mechanism (Eq. 12).

Weakness：
(1)	Particle-count constraint:  It requires ≥3 particles systems for stable Li prediction (Section 4.5), making it unsuitable for two-particle systems.
(2)	Generalizability: Experiments are limited to HEP tasks; potential applications in other domains remain untested.

---

> ### Author Rebuttal · Authors · 2025-07-30
>
> Thank you for the detailed review of our submission. We are glad that you value the clarity and the rigor of our work. Thanks as well for the questions and criticisms, which we address in the following.
>
> ### 1 Local frame prediction for $<3$ particles
>
> > It requires ≥3 particles systems for stable $L_i$ prediction (Section 4.5), making it unsuitable for two-particle systems. [...] For two-particle systems, could virtual particles or preprocessing be introduced?
>
> The local frame construction described in Sec. 4.5 relies on the equivariant prediction of three linearly independent 4-vectors $v_{i,k}$. In the case of one or two particles, there are at most two independent directions, the 4-momenta $p_1$ and $p_2$. In this very special case, it is not possible to construct equivariant local frames based on the information contained in the input point cloud. Linear combinations of the 4-momenta, like virtual particles, are not linearly independent and can therefore not solve the problem. In our code, we handle this special case by sampling the missing directions randomly from a $\mathrm{SO}(3)$-invariant distribution, at the cost of formally breaking Lorentz-equivariance to $\mathrm{SO}(3)$-equivariance.
>
> In situations with partial Lorentz symmetry breaking, the reference particles encoding the time and beam direction can lift the number of input vectors $p_i$ to 3 or higher. This happens in our jet tagging experiment, where 0.001% of the jets consist of only one or two particles.
>
> Machine learning tasks in high-energy physics where Lorentz symmetry is relevant typically feature large numbers of particles. We therefore think that the constraint to $\ge 3$ particle systems is no problem in practice.
>
> ### 2 Experiments beyond HEP tasks
>
> > Experiments are limited to HEP tasks; potential applications in other domains remain untested.
>
> The experiments intentionally focus on high‑energy physics, where precise benchmarks and strong Lorentz‑equivariant baselines allow us to show that local canonicalization with tensorial messages scales to demanding real‑world tasks. We agree that cross‑domain validation would be valuable and have added a broader evaluation to our future‑work plan.
>
> ### 3 Extension of our framework to other non-compact groups
>
> > Can the framework be modified to support computations for other non-compact groups (e.g., the Poincaré group)?
>
> In addition to Lorentz-equivariance, invariance w.r.t.~translations is typically obtained by using only differences of four-vectors. In all our experiments, the input is provided in momentum space where translation equivariance/invariance is not preferable. However, the extension to the Poincaré group is possible by modifying Eq. (13) to
>
> $$v_{i,k} = \sum_{j=1}^N\mathrm{softmax}(\varphi_k(s_i, s_j, \| x_i - x_j \|))(x_i - x_j) ,$$
>
> where we assume to work in position space with four-vectors $x_i$. One possible challenge may be to ensure that at least one of the predicted $v_{i,k}$ has a positive norm from which one can construct the boost $B$, cf. Alg. 1.
>
> Further, it is possible to directly extend our framework (including the tensorial message passing) to the group of $\mathrm O(1,d)$. We refer to the answer to reviewer VHeN for a broader discussion.
>
> We hope we were able to address your questions and look forward to discussing further.

---

### Official Review · Reviewer_sk9Q · 2025-07-03

**Clarity:** 3
**Significance:** 4
**Originality:** 3
**Rating:** 5
**Confidence:** 3

**Summary:**

This paper introduces LLoCa, a framework that makes any neural network design Lorentz-invariant for use in high-energy physics. The main idea is to use equivariantly predicted local reference frames to turn particle properties into Lorentz-invariant representations.

These representations can then be processed by traditional architectures like transformers or graph networks before being transformed back to produce equivariant outputs.

The authors incorporate the non-compact Lorentz group into geometric message passing and develop an attention mechanism based on the Minkowski product.

LLoCa is four times faster and uses 5 to 100 times fewer FLOPs than other Lorentz-equivariant methods, and it can even make vanilla transformers competitive without modifying their architecture for a specific domain.

The approach also enables fair comparison between precise equivariance and data augmentation, showing that exact Lorentz equivariance performs better with large amounts of training data.

**Questions:**

You mention that large boosts B may lead to numerical instabilities during orthonormalization, but that predicted vectors vi,k are typically not highly boosted. Can you provide a more rigorous analysis of when this stability breaks down?

Your baselines primarily compare against established methods. How does LLoCa perform against more recent equivariant architectures?

Provide more detailed algorithmic descriptions and pseudocode for the local reference frame construction, particularly the numerical procedures for ensuring stability.

**Ethical Concerns:**

["NO or VERY MINOR ethics concerns only"]

**Limitations:**

yes

**Paper Formatting Concerns:**

The paper's formatting is actually quite good overall. These are very minor issues that don't significantly detract from readability or comprehension.

**Quality:**

3

**Strengths And Weaknesses:**

**Strengths:** The core idea of using equivariantly predicted local reference frames to achieve Lorentz equivariance is both theoretically solid and practically valuable, as it separates equivariance from architectural limitations. The authors provide strong theoretical foundations with thorough proofs, demonstrate the framework's versatility by successfully applying it to various architectures (transformers, graph networks, domain-specific models), and present impressive empirical results that consistently outperform existing methods. The approach is also practically beneficial for large-scale applications because it substantially boosts computational efficiency—training is four times faster, and FLOPs are reduced by 5 to 100 times compared to specialized Lorentz-equivariant designs.

Furthermore, incorporating geometric message passing for the non-compact Lorentz group and developing Minkowski product-based attention are technically sophisticated contributions that enhance the field's theoretical understanding.

I have no more concerns.

---

> ### Author Rebuttal · Authors · 2025-07-30
>
> Thank you for the thorough and constructive review. We are happy to hear that you appreciate the theoretical soundness and practical value of our approach. Thanks as well for the questions and criticisms, which we address in the following.
>
> ### 1 Large boosts and numerical stability
>
> > You mention that large boosts B may lead to numerical instabilities during orthonormalization, but that predicted vectors $v_{i,k}$ are typically not highly boosted. Can you provide a more rigorous analysis of when this stability breaks down?
>
> Local frames with large boosts $B$ are problematic, because they multiply the latent features in each message passing step, possibly leading to exploding gradients. Such large boosts are caused by vectors $v_{i,0}$ constructed in Eq. (13) that have small but positive norm, because they yield large values for $\gamma$ in Eq. (2). These small-norm vectors $v_{i,0}$ can be caused by the typically small-norm particle 4-momenta $p_i$ in Eq. (13) either if all 4-momenta $p_i$ point in a similar direction, or if the coefficient $\mathrm{softmax}(\varphi_k (\dots))$ is small for all except one particle.
>
> Even bigger problems occur if the vectors $v_{i,0}$ have zero or negative norm, because the boost matrix $B$ in Eq. (2) is ill-defined due to $\vec \beta^2 \ge 1$. In practice, such pathological vectors $v_{i,0}$ do not occur with our implementation.
>
> We have identified several techniques to avoid large boosts already at initialization, where such instabilities are most likely to occur. First, we include a softmax operation in Eq. (13) to prevent any negative-norm vectors in the process. Second, we use regulator masses to enforce a lower limit on the norms of the input particles $p_i$. Third, we multiply by $p_i+p_j$ instead of simply $p_i$ in Eq. (13), because $p_i+p_j$ typically is not strongly boosted anymore.
>
> ### 2 Comparison against recent Lorentz-equivariant architectures
>
> > Your baselines primarily compare against established methods. How does LLoCa perform against more recent equivariant architectures?
>
> Throughout the paper we benchmark LLoCa against L-GATr [1], the most recent Lorentz-equivariant architecture. For jet tagging, L-GATr is the only Lorentz-equivariant architecture with public results on JetClass besides our LLoCa networks. We conducted an additional experiment by retraining LorentzNet [2] based on its official implementation, and using the same training hyperparameters as for LLoCa‑ParticleNet. LorentzNet matches LLoCa‑ParticleNet in accuracy but is ~30% slower. The transformers (L‑GATr and LLoCa‑Transformer) clearly outperform the graph networks, underscoring the scalability advantage of transformers.
>
> Network | Time | Accuracy | AUC
> --- | --- | --- | ---
> LLoCa-Transformer | 33h | 0.864 | 0.9882
> L-GATr | 180h | 0.866 | 0.9885
> LLoCa-ParticleNet | 41h | 0.848 | 0.9857
> LorentzNet (new) | 57h | 0.847 | 0.9856
>
> ### 3 Numerical details of local frame construction
>
> > Provide more detailed algorithmic descriptions and pseudocode for the local reference frame construction, particularly the numerical procedures for ensuring stability.
>
> Thanks for the suggestion. We will add a detailed algorithmic description in an appendix of the revised version of the paper. We will also publish our code upon publication, allowing the community to profit from our implementation of the local frame construction.
>
> First, most particles at the LHC can be considered massless as the typical energy scale is much larger than the mass of single particles. When operating on four-momenta using single precision, the particle mass value is often modified due to numerical underflows, leading to particles with zero mass in some cases. To avoid downstream instabilities, we enforce a minimal particle mass $m_\epsilon$ by increasing the energy of all input particles $p_i$ as $E' = \sqrt{m_\epsilon^2 + E^2}$, hence $m'^2 = m_\epsilon^2 + m^2$. In our experiments, we use $m_\epsilon=10^{-5}$ and $m_\epsilon=5\cdot 10^{-3}$ for the amplitude regression and tagging experiments, respectively.
>
> Second, we carefully design Eq. (13) to avoid vectors $v_{i,0}$ with small or negative norm which would cause instabilities due to large boosts. We discuss our design choices in the answer to the first question.
>
> Third, the $\mathrm{SO(3)}$ Gram-Schmidt orthonormalization used in Alg.1 requires two linearly independent vectors. We ensure that the two vectors $\vec w_1$, $\vec w_2$ are not collinear by calculating the distance between the two vectors. Concretely, if $(\vec v_1-\vec v_2)^2 < \epsilon_\text{collinear}^2$, we keep $\vec v_1$ and replace $\vec v_2$ by $\vec v_2 + \vec \epsilon$ with a random direction $\vec \epsilon$. In our experiments, we set $\epsilon_\text{collinear}=10^{-4}$ and we observe that, besides a handful of exceptions at initialization, the predicted vectors are never regularized. This indicates that our orthonormalization procedure is well under control.
>
> We hope we were able to address your questions and look forward to discussing further.
>
> **References**
>
> - [1] J. Spinner et al, "Lorentz-Equivariant Geometric Algebra Transformers for High-Energy Physics", NeurIPS 2024
> - [2] S. Gong et al, "An efficient Lorentz equivariant graph neural network for jet tagging", JHEP 2022
> - [3] A. Bogatskiy et al, "Explainable equivariant neural networks for particle physics: PELICAN", JHEP 2023
> - [4] D. Ruhe et al, "Clifford Group Equivariant Neural Networks", NeurIPS 2023

---

> > ### Comment · Reviewer_sk9Q · 2025-08-03
> >
> > Thanks for the author's detailed response. My concerns have been resolved.

---

> > > ### Author Response · Authors · 2025-08-07
> > >
> > > Many thanks for acknowledging our rebuttal. We are very glad that our answers have helped to resolve all of your concerns.

---

### Note · Authors · 2025-08-12

Once again, we would like to thank all reviewers for their time and thoughtful feedback. In summary, in response to all of the helpful comments, we have
* conducted several additional experiments, highlighting that our proposed framework significantly lifts the performance of non-equivariant backbone architectures even when using the same computational resources;
* laid out the generalization of the LLoCa framework to $\mathrm O(1,d)$ and the Poincaré group;
* provided additional details on the numerical stability of our local frame prediction;
* elaborated on the relation between LLoCa and existing works on canonicalization, in particular frame averaging.

We are very happy to hear that we were able to address the reviewers' questions adequately.

---

### Decision · Program_Chairs · 2025-09-17

**Decision:**

Accept (poster)

**Comment:**

This paper introduces Lorentz Local Canonicalization (LLoCa), a general framework that makes arbitrary backbone networks exactly Lorentz-equivariant by predicting local reference frames. The approach enables the construction of Lorentz-equivariant transformers and graph networks, extends geometric message passing to the Lorentz group, and naturally incorporates data augmentation as a frame choice. Empirical results show state-of-the-art accuracy on particle physics tasks with improved efficiency (faster inference and fewer FLOPs). Overall, the paper presents a novel, flexible, and effective method for advancing Lorentz-equivariant architectures in high-energy physics.

During the review and rebuttal period, all reviewers held a positive score for this paper, recognizing its novelty and technical contribution.
Therefore, I recommend the acceptance of this paper.